# RNA Secondary Structure Prediction By Learning Unrolled Algorithms

**Xinshi Chen**[1*]**, Yu Li**[2*]**, Ramzan Umarov**[2]**, Xin Gao**[2,†]**, Le Song**[1,3,†]
[1]Georgia Tech  [2]KAUST  [3]Ant Financial
xinshi.chen@gatech.edu
{yu.li;ramzan.umarov;xin.gao}@kaust.edu.sa
lsong@cc.gatech.edu

## ABSTRACT

In this paper, we propose an end-to-end deep learning model, called E2Efold, for RNA secondary structure prediction which can effectively take into account the inherent constraints in the problem. The key idea of E2Efold is to directly predict the RNA base-pairing matrix, and use an unrolled algorithm for constrained programming as the template for deep architectures to enforce constraints. With comprehensive experiments on benchmark datasets, we demonstrate the superior performance of E2Efold: it predicts significantly better structures compared to previous SOTA (especially for pseudoknotted structures), while being as efficient as the fastest algorithms in terms of inference time.

## 1 INTRODUCTION

Ribonucleic acid (RNA) is a molecule playing essential roles in numerous cellular processes and regulating expression of genes (Crick, 1970). It consists of an ordered sequence of nucleotides, with each nucleotide containing one of four bases: *Adenine (A), Guanine (G), Cytosine (C)* and *Uracile (U)*. This sequence of bases can be represented as

$$\boldsymbol{x} := (x_1, \ldots, x_L) \text{ where } x_i \in \{A, G, C, U\},$$

which is known as the *primary structure* of RNA. The bases can bond with one another to form a set of base-pairs, which defines the *secondary structure*. A secondary structure can be represented by a binary matrix $A^*$ where $A^*_{ij} = 1$ if the $i, j$-th

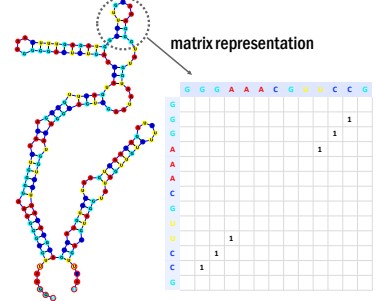

Figure 1: Graph and matrix representations of RNA secondary structure.

bases are paired (Fig 1). Discovering the secondary structure of RNA is important for understanding functions of RNA since the structure essentially affects the interaction and reaction between RNA and other cellular components. Although secondary structure can be determined by experimental assays (e.g. X-ray diffraction), it is slow, expensive and technically challenging. Therefore, computational prediction of RNA secondary structure becomes an important task in RNA research and is useful in many applications such as drug design (Iorns et al., 2007).

Research on computational prediction of RNA secondary structure from knowledge of primary structure has been carried out for decades. Most existing methods assume the secondary structure is a result of energy minimization, i.e., $A^* = \arg\min_A E_{\boldsymbol{x}}(A)$. The energy function is either estimated by physics-based thermodynamic experiments (Lorenz et al., 2011; Bellaousov et al., 2013; Markham & Zuker, 2008) or learned from data (Do et al.,

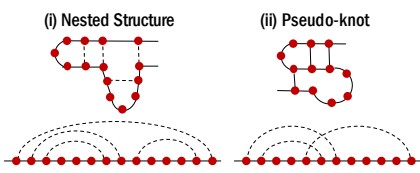

Figure 2: Nested and non-nested structures.

2006). These approaches are faced with a common problem that the search space of *all valid secondary structures* is exponentially-large with respect to the length $L$ of the sequence. To make the minimization tractable, it is often assumed the base-pairing has a *nested* structure (Fig 2 left), and

---

*Equal contribution. †Co-corresponding.

the energy function factorizes pairwisely. With this assumption, dynamic programming (DP) based algorithms can iteratively find the optimal structure for subsequences and thus consider an enormous number of structures in time $\mathcal{O}(L^3)$.

Although DP-based algorithms have dominated RNA structure prediction, it is notable that they restrict the search space to *nested structures*, which excludes some valid yet biologically important RNA secondary structures that contain '*pseudoknots*', i.e., elements with at least two non-nested base-pairs (Fig 2 right). Pseudoknots make up roughly 1.4% of base-pairs (Mathews & Turner, 2006), and are overrepresented in functionally important regions (Hajdin et al., 2013; Staple & Butcher, 2005). Furthermore, pseudoknots are present in around 40% of the RNAs. They also assist folding into 3D structures (Fechter et al., 2001) and thus should not be ignored. To predict RNA structures with pseudoknots, energy-based methods need to run more computationally intensive algorithms to decode the structures.

In summary, in the presence of more complex structured output (i.e., pseudoknots), it is challenging for energy-based approaches to simultaneously take into account the complex constraints while being efficient. In this paper, we adopt a different viewpoint by assuming that the secondary structure is the output of a feed-forward function, i.e., $A^* = \mathcal{F}_\theta(\boldsymbol{x})$, and propose to learn $\theta$ from data in an end-to-end fashion. It avoids the second minimization step needed in energy function based approach, and does not require the output structure to be nested. Furthermore, the feed-forward model can be fitted by directly optimizing the loss that one is interested in.

Despite the above advantages of using a feed-forward model, the architecture design is challenging. To be more concrete, in the RNA case, $\mathcal{F}_\theta$ is difficult to design for the following reasons:

(i) RNA secondary structure needs to obey certain *hard constraints* (see details in Section 3), which means certain kinds of pairings cannot occur at all (Steeg, 1993). Ideally, the output of $\mathcal{F}_\theta$ needs to satisfy these constraints.

(ii) The number of RNA data points is limited, so we cannot expect that a naive fully connected network can learn the predictive information and constraints directly from data. Thus, inductive biases need to be encoded into the network architecture.

(iii) One may take a two-step approach, where a post-processing step can be carried out to enforce the constraints when $\mathcal{F}_\theta$ predicts an invalid structure. However, in this design, the deep network trained in the first stage is unaware of the post-processing stage, making less effective use of the potential prior knowledge encoded in the constraints.

In this paper, we present an end-to-end deep learning solution which integrates the two stages. The first part of the architecture is a transformer-based deep model called *Deep Score Network* which represents sequence information useful for structure prediction. The second part is a multilayer network called *Post-Processing Network* which gradually enforces the constraints and restrict the output space. It is designed based on an unrolled algorithm for solving a constrained optimization. These two networks are coupled together and learned jointly in an end-to-end fashion. Therefore, we call our model **E2Efold**.

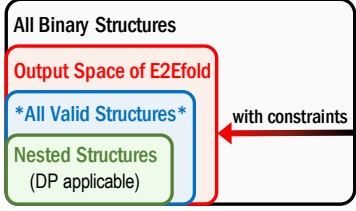

Figure 3: Output space of E2Efold.

By using an unrolled algorithm as the inductive bias to design Post-Processing Network, the output space of E2Efold is constrained (illustrated in Fig 3), which makes it easier to learn a good model in the case of limited data and also reduces the overfitting issue. Yet, the constraints encoded in E2Efold are flexible enough such that pseudoknots are not excluded. In summary, E2Efold strikes a nice balance between model biases for learning and expressiveness for valid RNA structures.

We conduct extensive experiments to compare E2Efold with state-of-the-art (SOTA) methods on several RNA benchmark datasets, showing superior performance of E2Efold including:

- being able to predict valid RNA secondary structures including pseudoknots;
- running as efficient as the fastest algorithm in terms of inference time;
- producing structures that are visually close to the true structure;
- better than previous SOTA in terms of F1 score, precision and recall.

Although in this paper we focus on RNA secondary structure prediction, which presents an important and concrete problem where E2Efold leads to significant improvements, our method is generic

and can be applied to other problems where constraints need to be enforced or prior knowledge is provided. We imagine that our design idea of learning unrolled algorithm to enforce constraints can also be transferred to problems such as protein folding and natural language understanding problems (e.g., building correspondence structure between different parts in a document).

## 2 RELATED WORK

**Classical RNA folding methods** identify candidate structures for an RNA sequence energy minimization through DP and rely on thousands of experimentally-measured thermodynamic parameters. A few widely used methods such as RNAstructure (Bellaousov et al., 2013), Vienna RNAfold (Lorenz et al., 2011) and UNAFold (Markham & Zuker, 2008) adpoted this approach. These methods typically scale as $\mathcal{O}(L^3)$ in time and $\mathcal{O}(L^2)$ in storage (Mathews, 2006), making them slow for long sequences. A recent advance called LinearFold (Huang et al., 2019) achieved linear run time $\mathcal{O}(L)$ by applying beam search, but it can not handle pseudoknots in RNA structures. The prediction of lowest free energy structures with pseudoknots is NP-complete (Lyngsø & Pedersen, 2000), so pseudoknots are not considered in most algorithms. Heuristic algorithms such as HotKnots (Andronescu et al., 2010) and Probknots (Bellaousov & Mathews, 2010) have been made to predict structures with pseudoknots, but the predictive accuracy and efficiency still need to be improved.

**Learning-based RNA folding methods** such as ContraFold (Do et al., 2006) and ContextFold (Zakov et al., 2011) have been proposed for energy parameters estimation due to the increasing availability of known RNA structures, resulting in higher prediction accuracies, but these methods still rely on the above DP-based algorithms for energy minimization. A recent deep learning model, CDPfold (Zhang et al., 2019), applied convolutional neural networks to predict base-pairings, but it adopts the dot-bracket representation for RNA secondary structure, which can not represent pseudoknotted structures. Moreover, it requires a DP-based post-processing step whose computational complexity is prohibitive for sequences longer than a few hundreds.

**Learning with differentiable algorithms** is a useful idea that inspires a series of works (Hershey et al., 2014; Belanger et al., 2017; Ingraham et al., 2018; Chen et al., 2018; Shrivastava et al., 2019), which shared similar idea of using differentiable unrolled algorithms as a building block in neural architectures. Some models are also applied to structured prediction problems (Hershey et al., 2014; Pillutla et al., 2018; Ingraham et al., 2018), but they did not consider the challenging RNA secondary structure problem or discuss how to properly incorporating constraints into the architecture. OptNet (Amos & Kolter, 2017) integrates constraints by differentiating KKT conditions, but it has cubic complexity in the number of variables and constraints, which is prohibitive for the RNA case.

**Dependency parsing in NLP** is a different but related problem to RNA folding. It predicts the dependency between the words in a sentence. Similar to nested/non-nested structures, the corresponding terms in NLP are projective/non-projective parsing, where most works focus on the former and DP-based inference algorithms are commonly used (McDonald et al., 2005). Deep learning models (Dozat & Manning, 2016; Kiperwasser & Goldberg, 2016) are proposed to proposed to score the dependency between words, which has a similar flavor to the Deep Score Network in our work.

## 3 RNA SECONDARY STRUCTURE PREDICTION PROBLEM

In the RNA secondary structure prediction problem, the input is the ordered sequence of bases $\boldsymbol{x} = (x_1, \ldots, x_L)$ and the output is the RNA secondary structure represented by a matrix $A^* \in \{0, 1\}^{L \times L}$. Hard constraints on the forming of an RNA secondary structure dictate that certain kinds of pairings cannot occur at all (Steeg, 1993). Formally, these constraints are:

| | | |
|---|---|---|
| (i) | Only three types of nucleotides combinations, $\mathcal{B} := \{AU, UA\} \cup \{GC, CG\} \cup \{GU, UG\}$, can form base-pairs. | $\forall i, j$, if $x_i x_j \notin \mathcal{B}$, then $A_{ij} = 0$. |
| (ii) | No sharp loops are allowed. | $\forall |i - j| < 4, A_{ij} = 0$. |
| (iii) | There is no overlap of pairs, i.e., it is a matching. | $\forall i, \sum_{j=1}^{L} A_{ij} \leq 1$. |

(i) and (ii) prevent pairing of certain base-pairs based on their types and relative locations. Incorporating these two constraints can help the model exclude lots of illegal pairs. (iii) is a global constraint among the entries of $A^*$.

The space of all valid secondary structures contains all *symmetric* matrices $A \in \{0, 1\}^{L \times L}$ that satisfy the above three constraints. This space is much smaller than the space of all binary matrices $\{0, 1\}^{L \times L}$. Therefore, if we could incorporate these constraints in our deep model, the reduced output space could help us train a better predictive model with less training data. We do this by using an unrolled algorithm as the inductive bias to design deep architecture.

## 4 E2EFOLD: DEEP LEARNING MODEL BASED ON UNROLLED ALGORITHM

In the literature on feed-forward networks for structured prediction, most models are designed using traditional deep learning architectures. However, for RNA secondary structure prediction, directly using these architectures does not work well due to the limited amount of RNA data points and the hard constraints on forming an RNA secondary structure. These challenges motivate the design of our E2Efold deep model, which combines a *Deep Score Network* with a *Post-Processing Network* based on an unrolled algorithm for solving a constrained optimization problem.

### 4.1 DEEP SCORE NETWORK

The first part of E2Efold is a *Deep Score Network* $U_\theta(\boldsymbol{x})$ whose output is an $L \times L$ symmetric matrix. Each entry of this matrix, i.e., $U_\theta(\boldsymbol{x})_{ij}$, indicates the score of nucleotides $x_i$ and $x_j$ being paired. The $\boldsymbol{x}$ input to the network here is the $L \times 4$ dimensional one-hot embedding. The specific architecture of $U_\theta$ is shown in Fig 4. It mainly consists of

- a position embedding matrix $\boldsymbol{P}$ which distinguishes $\{x_i\}_{i=1}^L$ by their exact and relative positions: $\boldsymbol{P}_i = \text{MLP}\big(\psi_1(i), \ldots, \psi_\ell(i), \psi_{\ell+1}(i/L), \ldots, \psi_n(i/L)\big)$, where $\{\psi_j\}$ is a set of $n$ feature maps such as $\sin(\cdot), \text{poly}(\cdot), \text{sigmoid}(\cdot)$, etc, and $\text{MLP}(\cdot)$ denotes multi-layer perceptions. Such position embedding idea has been used in natural language modeling such as BERT (Devlin et al., 2018), but we adapted for RNA sequence representation;
- a stack of Transformer Encoders (Vaswani et al., 2017) which encode the sequence information and the global dependency between nucleotides;
- a 2D Convolution layers (Wang et al., 2017) for outputting the pairwise scores.

With the representation power of neural networks, the hope is that we can learn an informative $U_\theta$ such that higher scoring entries in $U_\theta(\boldsymbol{x})$ correspond well to actual paired bases in RNA structure. Once the score matrix $U_\theta(\boldsymbol{x})$ is computed, a naive approach to use it is to choose an offset term $s \in \mathbb{R}$ (e.g., $s = 0$) and let $A_{ij} = 1$ if $U_\theta(\boldsymbol{x})_{ij} > s$. However, such entry-wise independent predictions of $A_{ij}$ may result in a matrix $A$ that violates the constraints for a valid RNA secondary structure. Therefore, a post-processing step is needed to make sure the predicted $A$ is valid. This step could be carried out separately after $U_\theta$ is learned. But such decoupling of base-pair scoring and post-processing for constraints may lead to sub-optimal results, where the errors in these two stages can not be considered together and tuned together. Instead, we will introduce a Post-Processing Network which can be trained end-to-end together with $U_\theta$ to enforce the constraints.

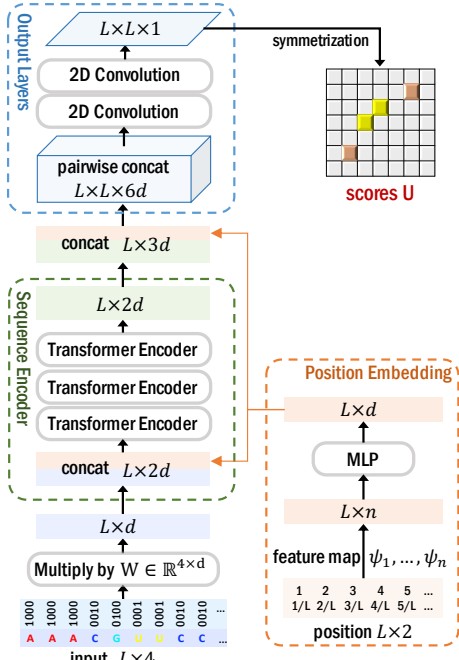

Figure 4: Architecture of *Deep Score Network*.

### 4.2 POST-PROCESSING NETWORK

The second part of E2Efold is a *Post-Processing Network* $\text{PP}_\phi$ which is an unrolled and parameterized algorithm for solving a constrained optimization problem. We first present how we formulate the post-processing step as a constrained optimization problem and the algorithm for solving it. After that, we show how we use the algorithm as a template to design deep architecture $\text{PP}_\phi$.

### 4.2.1 POST-PROCESSING WITH CONSTRAINED OPTIMIZATION

**Formulation of constrained optimization.** Given the scores predicted by $U_\theta(\boldsymbol{x})$, we define the total score $\frac{1}{2}\sum_{i,j}(U_\theta(\boldsymbol{x})_{ij} - s)A_{ij}$ as the objective to maximize, where $s$ is an offset term. Clearly, without structure constraints, the optimal solution is to take $A_{ij} = 1$ when $U_\theta(\boldsymbol{x})_{ij} > s$. Intuitively, the objective measures the covariation between the entries in the scoring matrix and the $A$ matrix. With constraints, the exact maximization becomes intractable. To make it tractable, we consider a convex relaxation of this discrete optimization to a continuous one by allowing $A_{ij} \in [0,1]$. Consequently, the solution space that we consider to optimize over is $\mathcal{A}(\boldsymbol{x}) := \left\{ A \in [0,1]^{L \times L} \mid A \text{ is symmetric and satisfies constraints (i)-(iii) in Section 3} \right\}$.

To further simplify the search space, we define a nonlinear transformation $\mathcal{T}$ on $\mathbb{R}^{L \times L}$ as $\mathcal{T}(\hat{A}) := \frac{1}{2}\left(\hat{A} \circ \hat{A} + (\hat{A} \circ \hat{A})^\top\right) \circ M(\boldsymbol{x})$, where $\circ$ denotes element-wise multiplication. Matrix $M$ is defined as $M(\boldsymbol{x})_{ij} := 1$ if $x_i x_j \in \mathcal{B}$ and also $|i - j| \geq 4$, and $M(\boldsymbol{x})_{ij} := 0$ otherwise. From this definition we can see that $M(\boldsymbol{x})$ encodes both constraint (i) and (ii). With transformation $\mathcal{T}$, the resulting matrix is non-negative, symmetric, and satisfies constraint (i) and (ii). Hence, by defining $A := \mathcal{T}(\hat{A})$, the solution space is simplified as $\mathcal{A}(\boldsymbol{x}) = \{A = \mathcal{T}(\hat{A}) \mid \hat{A} \in \mathbb{R}^{L \times L}, A\mathbf{1} \leq \mathbf{1}\}$.

Finally, we introduce a $\ell_1$ penalty term $\|\hat{A}\|_1 := \sum_{i,j}|\hat{A}_{ij}|$ to make $A$ sparse and formulate the post-processing step as: ($\langle \cdot, \cdot \rangle$ denotes matrix inner product, i.e., sum of entry-wise multiplication)

$$\max_{\hat{A} \in \mathbb{R}^{L \times L}} \frac{1}{2}\left\langle U_\theta(\boldsymbol{x}) - s, A := \mathcal{T}(\hat{A}) \right\rangle - \rho\|\hat{A}\|_1 \quad \text{s.t.} \ A\mathbf{1} \leq \mathbf{1} \tag{1}$$

The advantages of this formulation are that the variables $\hat{A}_{ij}$ are free variables in $\mathbb{R}$ and there are only $L$ inequality constraints $A\mathbf{1} \leq \mathbf{1}$. This system of linear inequalities can be replaced by a set of nonlinear equalities $\mathrm{relu}(A\mathbf{1} - \mathbf{1}) = \mathbf{0}$ so that the constrained problem can be easily transformed into an unconstrained problem by introducing a Lagrange multiplier $\boldsymbol{\lambda} \in \mathbb{R}_+^L$:

$$\min_{\boldsymbol{\lambda} \geq \mathbf{0}} \max_{\hat{A} \in \mathbb{R}^{L \times L}} \underbrace{\frac{1}{2}\langle U_\theta(\boldsymbol{x}) - s, A \rangle - \langle \boldsymbol{\lambda}, \mathrm{relu}(A\mathbf{1} - \mathbf{1}) \rangle}_{f} - \rho\|\hat{A}\|_1. \tag{2}$$

**Algorithm for solving it.** We use a primal-dual method for solving Eq. 2 (derived in Appendix B). In each iteration, $\hat{A}$ and $\boldsymbol{\lambda}$ are updated alternatively by:

(primal) gradient step: $\quad \dot{A}_{t+1} \leftarrow \hat{A}_t + \alpha \cdot \gamma_\alpha^t \cdot \hat{A}_t \circ M(\boldsymbol{x}) \circ \left(\partial f/\partial A_t + (\partial f/\partial A_t)^\top\right),$ (3)

$\qquad\qquad\qquad\quad$ where $\begin{cases} \partial f/\partial A_t = \frac{1}{2}(U_\theta(\boldsymbol{x}) - s) - (\boldsymbol{\lambda} \circ \mathrm{sign}(A_t\mathbf{1} - \mathbf{1}))\mathbf{1}^\top, \\ \mathrm{sign}(c) := 1 \text{ when } c > 0 \text{ and } 0 \text{ otherwise,} \end{cases}$ (4)

(primal) soft threshold: $\quad \hat{A}_{t+1} \leftarrow \mathrm{relu}(|\dot{A}_{t+1}| - \rho \cdot \alpha \cdot \gamma_\alpha^t), \quad A_{t+1} \leftarrow \mathcal{T}(\hat{A}_{t+1}),$ (5)

(dual) gradient step: $\quad \boldsymbol{\lambda}_{t+1} \leftarrow \boldsymbol{\lambda}_{t+1} + \beta \cdot \gamma_\beta^t \cdot \mathrm{relu}(A_{t+1}\mathbf{1} - \mathbf{1}),$ (6)

where $\alpha, \beta$ are step sizes and $\gamma_\alpha, \gamma_\beta$ are decaying coefficients. When it converges at $T$, an approximate solution $Round\left(A_T = \mathcal{T}(\hat{A}_T)\right)$ is obtained. With this algorithm operated on the learned $U_\theta(\boldsymbol{x})$, even if this step is disconnected to the training phase of $U_\theta(\boldsymbol{x})$, the final prediction works much better than many other existing methods (as reported in Section 6). Next, we introduce how to couple this post-processing step with the training of $U_\theta(\boldsymbol{x})$ to further improve the performance.

### 4.2.2 POST-PROCESSING NETWORK VIA AN UNROLLED ALGORITHM

We design a *Post-Processing Network*, denoted by $\mathrm{PP}_\phi$, based on the above algorithm. After it is defined, we can connect it with the deep score network $U_\theta$ and train them jointly in an end-to-end fashion, so that the training phase of $U_\theta(\boldsymbol{x})$ is aware of the post-processing step.

| **Algorithm 1:** Post-Processing Network $\text{PP}_\phi(U, M)$ | **Algorithm 2:** Neural Cell $\text{PPcell}_\phi$ |
|---|---|
| *Parameters* $\phi := \{w, s, \alpha, \beta, \gamma_\alpha, \gamma_\beta, \rho\}$ | **Function** $\texttt{PPcell}_\phi\,(U, M, \boldsymbol{\lambda}, A, \hat{A}, t)$**:** |
| $U \leftarrow \text{softsign}(U - s) \circ U$ | $\quad G \leftarrow \frac{1}{2} U - (\boldsymbol{\lambda} \circ \text{softsign}(A\mathbf{1} - \mathbf{1}))\,\mathbf{1}^\top$ |
| $\hat{A}_0 \leftarrow \text{softsign}(U - s) \circ \text{sigmoid}(U)$ | $\quad \dot{A} \leftarrow \hat{A} + \alpha \cdot \gamma_\alpha{}^t \cdot \hat{A} \circ M \circ (G + G^\top)$ |
| $A_0 \leftarrow \mathcal{T}(\hat{A}_0); \quad \boldsymbol{\lambda}_0 \leftarrow w \cdot \text{relu}(A_0\mathbf{1} - \mathbf{1})$ | $\quad \hat{A} \leftarrow \text{relu}(\lvert\dot{A}\rvert - \rho \cdot \alpha \cdot \gamma_\alpha{}^t)$ |
| **For** $t = 0, \ldots, T - 1$ **do** | $\quad \hat{A} \leftarrow 1 - \text{relu}(1 - \hat{A}) \quad [\text{i.e.,}\min(\hat{A}, 1)]$ |
| $\quad \boldsymbol{\lambda}_{t+1}, A_{t+1}, \hat{A}_{t+1} = \text{PPcell}_\phi(U, M, \boldsymbol{\lambda}_t, A_t, \hat{A}_t, t)$ | $\quad A \leftarrow \mathcal{T}(\hat{A}); \boldsymbol{\lambda} \leftarrow \boldsymbol{\lambda} + \beta \cdot \gamma_\beta{}^t \cdot \text{relu}(A\mathbf{1} - \mathbf{1})$ |
| **return** $\{A_t\}_{t=1}^T$ | $\quad$**return** $\boldsymbol{\lambda}, A, \hat{A}$ |

The specific computation graph of $\text{PP}_\phi$ is given in Algorithm 1, whose main component is a recurrent cell which we call $\text{PPcell}_\phi$. The computation graph is almost the same as the iterative update from Eq. 3 to Eq. 6, except for several modifications:

- *(learnable hyperparameters)* The hyperparameters including step sizes $\alpha, \beta$, decaying rate $\gamma_\alpha, \gamma_\beta$, sparsity coefficient $\rho$ and the offset term $s$ are treated as learnable parameters in $\phi$, so that there is no need to tune the hyperparameters by hand but automatically learn them from data instead.
- *(fixed # iterations)* Instead of running the iterative updates until convergence, $\text{PPcell}_\phi$ is applied recursively for $T$ iterations where $T$ is a manually fixed number. This is why in Fig 3 the output space of E2Efold is slightly larger than the true solution space.
- *(smoothed sign function)* Resulted from the gradient of $\text{relu}(\cdot)$, the update step in Eq. 4 contains a $\text{sign}(\cdot)$ function. However, to push gradient through $\text{PP}_\phi$, we require a differentiable update step. Therefore, we use a smoothed sign function defined as $\text{softsign}(c) := 1/(1 + \exp(-kc))$, where $k$ is a temperature.
- *(clip $\hat{A}$)* An additional step, $\hat{A} \leftarrow \min(\hat{A}, 1)$, is included to make the output $A_t$ at each iteration stay in the range $[0, 1]^{L \times L}$. This is useful for computing the loss over intermediate results $\{A_t\}_{t=1}^T$, for which we will explain more in Section 5.

With these modifications, the Post-Processing Network $\text{PP}_\phi$ is a tuning-free and differentiable unrolled algorithm with meaningful intermediate outputs. Combining it with the deep score network, the final deep model is

$$\textbf{E2Efold}: \quad \{A_t\}_{t=1}^T = \overbrace{\text{PP}_\phi(\ \underbrace{U_\theta(\boldsymbol{x})}_{\textit{Deep Score Network}}\ , M(\boldsymbol{x}))}^{\textit{Post-Process Network}}. \tag{7}$$

## 5 END-TO-END TRAINING ALGORITHM

Given a dataset $\mathcal{D}$ containing examples of input-output pairs $(\boldsymbol{x}, A^*)$, the training procedure of E2Efold is similar to standard gradient-based supervised learning. However, for RNA secondary structure prediction problems, commonly used metrics for evaluating predictive performances are F1 score, precision and recall, which are non-differentiable.

**Differentiable F1 Loss.** To directly optimize these metrics, we mimic true positive (TP), false positive (FP), true negative (TN) and false negative (FN) by defining continuous functions on $[0, 1]^{L \times L}$:

$$\text{TP} = \langle A, A^* \rangle,\ \text{FP} = \langle A, 1 - A^* \rangle,\ \text{FN} = \langle 1 - A, A^* \rangle,\ \text{TN} = \langle 1 - A, 1 - A^* \rangle.$$

Since $\text{F1} = 2\text{TP}/(2\text{TP} + \text{FP} + \text{FN})$, we define a loss function to mimic the negative of F1 score as:

$$\mathcal{L}_{-\text{F1}}(A, A^*) := -2\langle A, A^* \rangle / \left( 2\langle A, A^* \rangle + \langle A, 1 - A^* \rangle + \langle 1 - A, A^* \rangle \right). \tag{8}$$

Assuming that $\sum_{ij} A_{ij}^* \neq 0$, this loss is well-defined and differentiable on $[0, 1]^{L \times L}$. Precision and recall losses can be defined in a similar way, but we optimize F1 score in this paper.

It is notable that this F1 loss takes advantages over other differentiable losses including $\ell_2$ and cross-entropy losses, because there are much more negative samples (i.e. $A_{ij} = 0$) than positive samples (i.e. $A_{ij} = 1$). A hand-tuned weight is needed to balance them while using $\ell_2$ or cross-entropy losses, but F1 loss handles this issue automatically, which can be useful for a number of problems (Wang et al., 2016; Li et al., 2017).

**Overall Loss Function.** As noted earlier, E2Efold outputs a matrix $A_t \in [0,1]^{L \times L}$ in each iteration. This allows us to add auxiliary losses to regularize the intermediate results, guiding it to learn parameters which can generate a smooth solution trajectory. More specifically, we use an objective that depends on the entire trajectory of optimization:

$$\min_{\theta,\phi} \frac{1}{|\mathcal{D}|} \sum_{(x,A^*)\in\mathcal{D}} \frac{1}{T} \sum_{t=1}^{T} \gamma^{T-t} \mathcal{L}_{-\mathrm{F1}}(A_t, A^*), \tag{9}$$

where $\{A_t\}_{t=1}^{T} = \mathrm{PP}_\phi(U_\theta(\boldsymbol{x}), M(\boldsymbol{x}))$ and $\gamma \leq 1$ is a discounting factor. Empirically, we find it very useful to pre-train $U_\theta$ using logistic regression loss. Also, it is helpful to add this additional loss to Eq. 9 as a regularization.

## 6 EXPERIMENTS

We compare E2Efold with the SOTA and also the most commonly used methods in the RNA secondary structure prediction field on two benchmark datasets. It is revealed from the experimental results that E2Efold achieves 29.7% improvement in terms of F1 score on RNAstralign dataset and it infers the RNA secondary structure as fast as the most efficient algorithm (LinearFold) among existing ones. An ablation study is also conducted to show the necessity of pushing gradient through the post-processing step. The codes for reproducing the experimental results are released.[1]

**Dataset.** We use two benchmark datasets: (i) ArchiveII (Sloma & Mathews, 2016), containing 3975 RNA structures from 10 RNA types, is a widely used benchmark dataset for classical RNA folding methods. (ii) RNAStralign (Tan et al., 2017), composed of 37149 structures from 8 RNA types, is one of the most comprehensive collections of RNA structures in the market. After removing redundant sequences and structures, 30451 structures remain. See Table 1 for statistics about these two datasets.

Table 1: Dataset Statistics

| Type | ArchiveII | | RNAStralign | |
|---|---|---|---|---|
| | length | #samples | length | #samples |
| All | 28~2968 | 3975 | 30~1851 | 30451 |
| 16SrRNA | 73~1995 | 110 | 54~1851 | 11620 |
| 5SrRNA | 102~135 | 1283 | 104~132 | 9385 |
| tRNA | 54~93 | 557 | 59~95 | 6443 |
| grp1 | 210~736 | 98 | 163~615 | 1502 |
| SRP | 28~533 | 928 | 30~553 | 468 |
| tmRNA | 102~437 | 462 | 102~437 | 572 |
| RNaseP | 120~486 | 454 | 189~486 | 434 |
| telomerase | 382~559 | 37 | 382~559 | 37 |
| 23SrRNA | 242~2968 | 35 | - | - |
| grp2 | 619~780 | 11 | - | - |

**Experiments On RNAStralign.** We divide RNAStralign dataset into training, testing and validation sets by stratified sampling (see details in Table 7 and Fig 6), so that each set contains all RNA types. We compare the performance of E2Efold to six methods including CDPfold, LinearFold, Mfold, RNAstructure (ProbKnot), RNAfold and CONTRAfold. Both E2Efold and CDPfold are learned from the same training/validation sets. For other methods, we directly use the provided packages or web-servers to generate predicted structures. We evaluate the F1 score, Precision and Recall for each sequence in the test set. Averaged values are reported in Table 2. As suggested by Mathews (2019), for a base pair $(i, j)$, the following predictions are also considered as correct: $(i+1, j), (i-1, j), (i, j+1), (i, j-1)$, so we also reported the metrics when one-position shift is allowed.

Table 2: Results on RNAStralign test set. "(S)" indicates the results when one-position shift is allowed.

| Method | Prec | Rec | F1 | Prec(S) | Rec(S) | F1(S) |
|---|---|---|---|---|---|---|
| **E2Efold** | **0.866** | **0.788** | **0.821** | **0.880** | **0.798** | **0.833** |
| CDPfold | 0.633 | 0.597 | 0.614 | 0.720 | 0.677 | 0.697 |
| LinearFold | 0.620 | 0.606 | 0.609 | 0.635 | 0.622 | 0.624 |
| Mfold | 0.450 | 0.398 | 0.420 | 0.463 | 0.409 | 0.433 |
| RNAstructure | 0.537 | 0.568 | 0.550 | 0.559 | 0.592 | 0.573 |
| RNAfold | 0.516 | 0.568 | 0.540 | 0.533 | 0.587 | 0.558 |
| CONTRAfold | 0.608 | 0.663 | 0.633 | 0.624 | 0.681 | 0.650 |

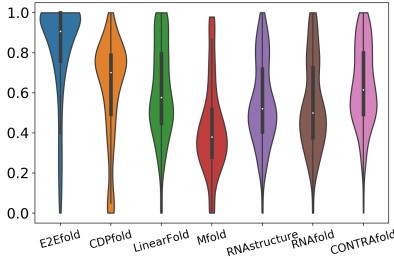

Figure 5: Distribution of F1 score.

As shown in Table 2, traditional methods can achieve a F1 score ranging from 0.433 to 0.624, which is consistent with the performance reported with their original papers. The two learning-based methods, CONTRAfold and CDPfold, can outperform classical methods with reasonable margin on

---

[1]The codes for reproducing the experimental results are released at https://github.com/ml4bio/e2efold.

some criteria. E2Efold, on the other hand, significantly outperforms all previous methods across all criteria, with at least 20% improvement. Notice that, for almost all the other methods, the recall is usually higher than precision, while for E2Efold, the precision is higher than recall. That can be the result of incorporating constraints during neural network training. Fig 5 shows the distributions of F1 scores for each method. It suggests that E2Efold has consistently good performance.

To estimate the performance of E2Efold on long sequences, we also compute the F1 scores weighted by the length of sequences, such that the results are more dominated by longer sequences. Detailed results are given in Appendix D.3.

Table 3: Performance comparison on ArchiveII

| Method | Prec | Rec | F1 | Prec(S) | Rec(S) | F1(S) |
|---|---|---|---|---|---|---|
| **E2Efold** | **0.734** | **0.66** | **0.686** | **0.758** | **0.676** | **0.704** |
| CDPfold | 0.557 | 0.535 | 0.545 | 0.612 | 0.585 | 0.597 |
| LinearFold | 0.641 | 0.617 | 0.621 | 0.668 | 0.644 | 0.647 |
| Mfold | 0.428 | 0.383 | 0.401 | 0.450 | 0.403 | 0.421 |
| RNAstructure | 0.563 | 0.615 | 0.585 | 0.590 | 0.645 | 0.613 |
| RNAfold | 0.565 | 0.627 | 0.592 | 0.586 | 0.652 | 0.615 |
| CONTRAfold | 0.607 | 0.679 | 0.638 | 0.629 | 0.705 | 0.662 |

Table 4: Inference time on RNAStralign

| Method | | total run time | time per seq |
|---|---|---|---|
| **E2Efold (Pytorch)** | | **19m (GPU)** | **0.40s** |
| CDPfold (Pytorch) | | 440m*32 threads | 300.107s |
| LinearFold | (C) | 20m | 0.43s |
| Mfold | (C) | 360m | 7.65s |
| RNAstructure | (C) | 3 days | 142.02s |
| RNAfold | (C) | 26m | 0.55s |
| CONTRAfold | (C) | 1 day | 30.58s |

**Test On ArchiveII Without Re-training.** To mimic the real world scenario where the users want to predict newly discovered RNA's structures which may have a distribution different from the training dataset, we directly test the model learned from RNAStralign training set on the ArchiveII dataset, without re-training the model. To make the comparison fair, we exclude sequences that are over-lapped with the RNAStralign dataset. We then test the model on sequences in ArchiveII that have overlapping RNA types (5SrRNA, 16SrRNA, etc) with the RNAStralign dataset. Results are shown in Table 3. It is understandable that the performances of classical methods which are not learning-based are consistent with that on RNAStralign. The performance of E2Efold, though is not as good as that on RNAStralign, is still better than all the other methods across different evaluation criteria. In addition, since the original ArchiveII dataset contains domain sequences (subsequences), we remove the domains and report the results in Appendix D.4, which are similar to results in Table 3.

**Inference Time Comparison.** We record the running time of all algorithms for predicting RNA secondary structures on the RNAStralign test set, which is summarized in Table 4. LinearFold is the most efficient among baselines because it uses beam pruning heuristic to accelerate DP. CDPfold, which achieves higher F1 score than other baselines, however, is extremely slow due to its DP post-processing step. Since we use a gradient-based algorithm which is simple to design the Post-Processing Network, E2Efold is fast. On GPU, E2Efold has similar inference time as LinearFold.

**Pseudoknot Prediction.** Even though E2Efold does not exclude pseudoknots, it is not sure whether it actually generates pseudoknotted structures. Therefore, we pick all sequences containing pseudoknots and compute the averaged F1 score only on this set. Besides, we count the number of pseudoknotted sequences that are

Table 5: Evaluation of pseudoknot prediction

| Method | Set F1 | TP | FP | TN | FN |
|---|---|---|---|---|---|
| E2Efold | 0.710 | 1312 | 242 | 1271 | 0 |
| RNAstructure | 0.472 | 1248 | 307 | 983 | 286 |

predicted as pseudoknotted and report this count as true positive (TP). Similarly we report TN, FP and FN in Table 5 along with the F1 score. Most tools exclude pseudoknots while RNAstructure is the most famous one that can predict pseudoknots, so we choose it for comparison.

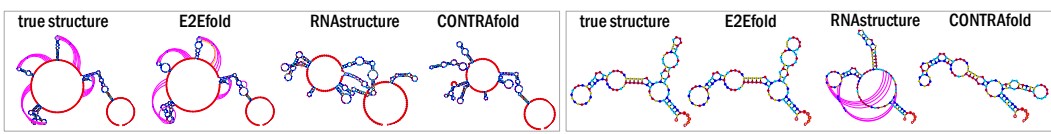

**Visualization.** We visualize predicted structures of three RNA sequences in the main text. More examples are provided in appendix (Fig 8 to 14). In these figures, purple lines indicate edges of pseudoknotted elements. Although CDPfold has higher F1 score than other baselines, its predictions are visually far from the ground-truth. Instead, RNAstructure and CONTRAfold produce

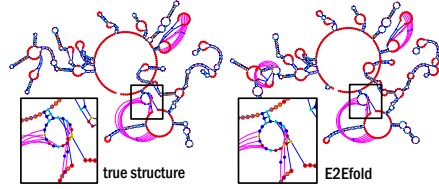

comparatively more reasonable visualizations among all baselines, so we compare with them. These

two methods can capture a rough sketch of the structure, but not good enough. For most cases, E2Efold produces structures most similar to the ground-truths. Moreover, it works surprisingly well for some RNA sequences that are long and very difficult to predict.

**Ablation Study.** To exam whether integrating the two stages by pushing gradient through the post-process is necessary for performance of E2Efold, we conduct an ablation study (Table 6). We test the performance when the post-processing step is discon-

Table 6: Ablation study (RNAStralign test set)

| Method | Prec | Rec | F1 | Prec(S) | Rec(S) | F1(S) |
|---|---|---|---|---|---|---|
| **E2Efold** | **0.866** | **0.788** | **0.821** | **0.880** | **0.798** | **0.833** |
| $U_\theta$+PP | 0.755 | 0.712 | 0.721 | 0.782 | 0.737 | 0.752 |

nected with the training of Deep Score Network $U_\theta$. We apply the post-processing step (i.e., for solving augmented Lagrangian) after $U_\theta$ is learned (thus the notation "$U_\theta$ + PP" in Table 6). Although "$U_\theta$ + PP" performs decently well, with constraints incorporated into training, E2Efold still has significant advantages over it.

**Discussion.** To better estimate the performance of E2Efold on different RNA types, we include the per-family F1 scores in Appendix D.5. E2Efold performs significantly better than other methods in 16S rRNA, tRNA, 5S RNA, tmRNA, and telomerase. These results are from a single model. In the future, we can view it as multi-task learning and further improve the performance by learning multiple models for different RNA families and learning an additional classifier to predict which model to use for the input sequence.

## 7 CONCLUSION

We propose a novel DL model, E2Efold, for RNA secondary structure prediction, which incorporates hard constraints in its architecture design. Comprehensive experiments are conducted to show the superior performance of E2Efold, no matter on quantitative criteria, running time, or visualization. Further studies need to be conducted to deal with the RNA types with less samples. Finally, we believe the idea of unrolling constrained programming and pushing gradient through post-processing can be generic and useful for other constrained structured prediction problems.

## ACKNOWLEDGEMENT

We would like to thank anonymous reviewers for providing constructive feedbacks. This work is supported in part by NSF grants CDS&E-1900017 D3SC, CCF-1836936 FMitF, IIS-1841351, CAREER IIS-1350983 to L.S. and grants from King Abdullah University of Science and Technology, under award numbers BAS/1/1624-01, FCC/1/1976-18-01, FCC/1/1976-23-01, FCC/1/1976-25-01, FCC/1/1976-26-01, REI/1/0018-01-01, and URF/1/4098-01-01.

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

## A  MORE DISCUSSION ON RELATED WORKS

Here we explain the difference between our approach and other works on unrolling optimization problems.

First, our view of incorporating constraints to reduce output space and to reduce sample complexity is novel. Previous works (Hershey et al., 2014; Belanger et al., 2017; Ingraham et al., 2018) did not discuss these aspects. The most related work which also integrates constraints is OptNet (Amos & Kolter, 2017), but its very expensive and can not scale to the RNA problem. Therefore, our proposed approach is a simple and effective one.

Second, compared to (Chen et al., 2018; Shrivastava et al., 2019), our approach has a different purpose of using the algorithm. Their goal is to learn a better algorithm, so they commonly make their architecture **more flexible** than the original algorithm for the room of improvement. However, we aim at enforcing constraints. To ensure that constraints are nicely incorporated, we **keep** the original structure of the algorithm and only make the hyperparameters learnable.

Finally, although all works consider end-to-end training, none of them can directly optimize the F1 score. We proposed a differentiable loss function to mimic the F1 score/precision/recall, which is effective and also very useful when negative samples are much fewer than positive samples (or the inverse).

## B  DERIVATION OF THE PROXIMAL GRADIENT STEP

The maximization step in Eq. 1 can be written as the following minimization:

$$\min_{\hat{A}\in\mathbb{R}^{L\times L}} \underbrace{-\frac{1}{2}\langle U_\theta(\boldsymbol{x})-s, A\rangle + \langle\boldsymbol{\lambda}, \mathrm{relu}(A\mathbf{1}-\mathbf{1})\rangle}_{-f(\hat{A})} +\rho\|\hat{A}\|_1. \tag{10}$$

Consider the quadratic approximation of $-f(\hat{A})$ centered at $\hat{A}_t$:

$$-\tilde{f}_\alpha(\hat{A}) := -f(\hat{A}_t) + \langle-\frac{\partial f}{\partial\hat{A}_t}, \hat{A}-\hat{A}_t\rangle + \frac{1}{2\alpha}\|\hat{A}-\hat{A}_t\|_F^2 \tag{11}$$

$$= -f(\hat{A}_t) + \frac{1}{2\alpha}\left\|\hat{A}-\left(\hat{A}_t+\alpha\frac{\partial f}{\partial\hat{A}_t}\right)\right\|_F^2, \tag{12}$$

and rewrite the optimization in Eq. 10 as

$$\min_{\hat{A}\in\mathbb{R}^{L\times L}} -f(\hat{A}_t) + \frac{1}{2\alpha}\left\|\hat{A}-\dot{A}_{t+1}\right\|_F^2 + \rho\|\hat{A}\|_1 \tag{13}$$

$$\equiv \min_{\hat{A}\in\mathbb{R}^{L\times L}} \frac{1}{2\alpha}\left\|\hat{A}-\dot{A}_{t+1}\right\|_F^2 + \rho\|\hat{A}\|_1, \tag{14}$$

where

$$\dot{A}_{t+1} := \hat{A}_t + \alpha\frac{\partial f}{\partial\hat{A}_t}. \tag{15}$$

Next, we define proximal mapping as a function depending on $\alpha$ as follows:

$$prox_\alpha(\dot{A}_{t+1}) = \arg\min_{\hat{A}\in\mathbb{R}^{L\times L}} \frac{1}{2\alpha}\left\|\hat{A}-\dot{A}_{t+1}\right\|_F^2 + \rho\|\hat{A}\|_1 \tag{16}$$

$$= \arg\min_{\hat{A}\in\mathbb{R}^{L\times L}} \frac{1}{2}\left\|\hat{A}-\dot{A}_{t+1}\right\|_F^2 + \alpha\rho\|\hat{A}\|_1 \tag{17}$$

$$= \mathrm{sign}(\dot{A}_{t+1})\max(|\dot{A}_{t+1}|-\alpha\rho, 0) \tag{18}$$

$$= \mathrm{sign}(\dot{A}_{t+1})\mathrm{relu}(|\dot{A}_{t+1}|-\alpha\rho). \tag{19}$$

Since we always use $\hat{A}\circ\hat{A}$ instead of $\hat{A}$ in our problem, we can take the absolute value $|prox_\alpha(\dot{A}_{t+1})| = \mathrm{relu}(|\dot{A}_{t+1}|-\alpha\rho)$ without loss of generality. Therefore, the proximal gradient

step is

$$\dot{A}_{t+1} \leftarrow \hat{A}_t + \alpha \frac{\partial f}{\partial \hat{A}_t} \quad \text{(correspond to Eq. 3)} \tag{20}$$

$$\hat{A}_{t+1} \leftarrow \text{relu}(|\dot{A}_{t+1}| - \alpha\rho) \quad \text{(correspond to Eq. 5).} \tag{21}$$

More specifically, in the main text, we write $\frac{\partial f}{\partial \hat{A}_t}$ as

$$\frac{\partial f}{\partial \hat{A}_t} = \frac{1}{2} \left( \frac{\partial f}{\partial A_t} + \frac{\partial f}{\partial A_t}^{\top} \right) \circ \frac{\partial A_t}{\partial \hat{A}_t} \tag{22}$$

$$= \left( \frac{1}{2} \frac{\partial A_t}{\partial \hat{A}_t} \right) \circ \left( \frac{\partial f}{\partial A_t} + \frac{\partial f}{\partial A_t}^{\top} \right) \tag{23}$$

$$= \left( \frac{1}{2^2} \circ M \circ (2\hat{A}_t + 2\hat{A}_t^{\top}) \right) \circ \left( \frac{\partial f}{\partial A_t} + \frac{\partial f}{\partial A_t}^{\top} \right) \tag{24}$$

$$= \left( \frac{1}{2^2} \circ M \circ (2\hat{A}_t + 2\hat{A}_t^{\top}) \right) \circ \left( \frac{\partial f}{\partial A_t} + \frac{\partial f}{\partial A_t}^{\top} \right) \tag{25}$$

$$= M \circ \hat{A}_t \circ \left( \frac{\partial f}{\partial A_t} + \frac{\partial f}{\partial A_t}^{\top} \right). \tag{26}$$

The last equation holds since $\hat{A}_t$ will remain symmetric in our algorithm if the initial $\hat{A}_0$ is symmetric. Moreover, in the main text, $\alpha$ is replaced by $\alpha \cdot \gamma_\alpha^t$.

## C   IMPLEMENTATION AND TRAINING DETAILS

We used Pytorch to implement the whole package of E2Efold.

**Deep Score Network.** In the deep score network, we used a hyper-parameter, $d$, which was set as 10 in the final model, to control the model capacity. In the transformer encoder layers, we set the number of heads as 2, the dimension of the feed-forward network as 2048, the dropout rate as 0.1. As for the position encoding, we used 58 base functions to form the position feature map, which goes through a 3-layer fully-connected neural network (the number of hidden neurons is $5 * d$) to generate the final position embedding, whose dimension is $L$ by $d$. In the final output layer, the pairwise concatenation is carried out in the following way: Let $X \in \mathbb{R}^{L \times 3d}$ be the input to the final output layers in Figure 4 (which is the concatenation of the sequence embedding and position embedding). The pairwise concatenation results in a tensor $Y \in \mathbb{R}^{L \times L \times 6d}$ defined as

$$Y(i, j, :) = [X(i, :), X(j, :)], \tag{27}$$

where $Y(i, j, :) \in \mathbb{R}^{6d}$, $X(i, :) \in \mathbb{R}^{3d}$, and $X(j, :) \in \mathbb{R}^{3d}$.

In the 2D convolution layers, the the channel of the feature map gradually change from $6 * d$ to $d$, and finally to 1. We set the kernel size as 1 to translate the feature map into the final score matrix. Each 2D convolution layer is followed by a batch normalization layer. We used ReLU as the activation function within the whole score network.

**Post-Processing Network.** In the PP network, we initialized $w$ as 1, $s$ as $\log(9)$, $\alpha$ as 0.01, $\beta$ as 0.1, $\gamma_\alpha$ as 0.99, $\gamma_\beta$ as 0.99, and $\rho$ as 1. We set $T$ as 20.

**Training details.** During training, we first pre-trained a deep score network and then fine-tuned the score network and the PP network together. To pre-train the score network, we used binary cross-entropy loss and Adam optimizer. Since, in the contact map, most entries are 0, we used weighted loss and set the positive sample weight as 300. The batch size was set to fully use the GPU memory, which was 20 for the Titan Xp card. We pre-train the score network for 100 epochs. As for the fine-tuning, we used binary cross-entropy loss for the score network and F1 loss for the PP network and summed up these two losses as the final loss. The user can also choose to only use the F1 loss or

use another coefficient to weight the loss estimated on the score network $U_\theta$. Due to the limitation of the GPU memory, we set the batch size as 8. However, we updated the model's parameters every 30 steps to stabilize the training process. We fine-tuned the whole model for 20 epochs. Also, since the data for different RNA families are imbalanced, we up-sampled the data in the small RNA families based on their size. For the training of the score network $U_\theta$ in the ablation study, it is exactly the same as the training of the above mentioned process. Except that during the fine-tune process, there is the unrolled number of iterations is set to be 0.

## D    MORE EXPERIMENTAL DETAILS

### D.1    DATASET STATISTICS

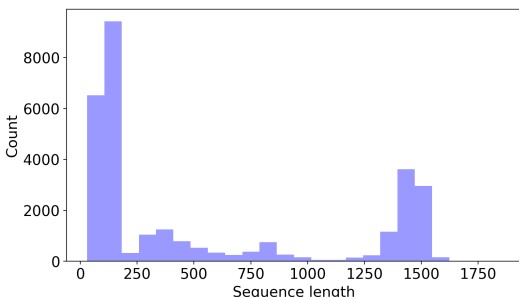

Figure 6: The RNAStralign length distribution.

Table 7: RNAStralign dataset splits statistics

| RNA type | All | Training | Validation | Testing |
| --- | --- | --- | --- | --- |
| 16SrRNA | 11620 | 9325 | 1145 | 1150 |
| 5SrRNA | 9385 | 7687 | 819 | 879 |
| tRNA | 6443 | 5412 | 527 | 504 |
| grp1 | 1502 | 1243 | 123 | 136 |
| SRP | 468 | 379 | 36 | 53 |
| tmRNA | 572 | 461 | 50 | 61 |
| RNaseP | 434 | 360 | 37 | 37 |
| telomerase | 37 | 28 | 4 | 5 |
| RNAStralign | 30451 | 24895 | 2702 | 2854 |

### D.2    TWO-SAMPLE HYPOTHESIS TESTING

To better understand the data distribution in different datasets, we provide statistical hypothesis test results in this section.

We can assume that

(i)  Samples in RNAStralign training set are i.i.d. from the distribution $\mathcal{P}(\text{RNAStr}_{\text{train}})$;

(ii)  Samples in RNAStralign testing set are i.i.d. from the distribution $\mathcal{P}(\text{RNAStr}_{\text{test}})$;

(iii)  Samples in ArchiveII dataset are i.i.d. from the distribution $\mathcal{P}(\text{ArcII})$.

To compare the differences among these data distributions, we can test the following hypothesis:

(a)  $\mathcal{P}(\text{RNAStr}_{\text{train}}) = \mathcal{P}(\text{RNAStr}_{\text{test}})$

(b)  $\mathcal{P}(\text{RNAStr}_{\text{train}}) = \mathcal{P}(\text{ArchiveII})$

The approach that we adopted is the permutation test on the unbiased empirical Maximum Mean Discrepancy (MMD) estimator:

$$\text{MMD}_u(X, Y) := \Big( \sum_{i=1}^{N} \sum_{j \neq i}^{N} k(x_i, x_j) + \sum_{i=1}^{M} \sum_{j \neq i}^{M} k(y_i, y_j) - \frac{2}{mn} \sum_{i=1}^{N} \sum_{j=1}^{M} k(x_i, y_j) \Big)^{\frac{1}{2}}, \qquad (28)$$

where $X = \{x_i\}_{i=1}^{N}$ contains $N$ i.i.d. samples from a distribution $\mathcal{P}_1$, $Y = \{y_i\}_{i=1}^{M}$ contains $M$ i.i.d. samples from a distribution $\mathcal{P}_2$, and $k(\cdot, \cdot)$ is a string kernel.

Since we conduct stratified sampling to split the training and testing dataset, when we perform permutation test, we use stratified re-sampling as well (for both Hypothese (a) and (b)). The result of the permutation test (permuted 1000 times) is reported in Figure 7.

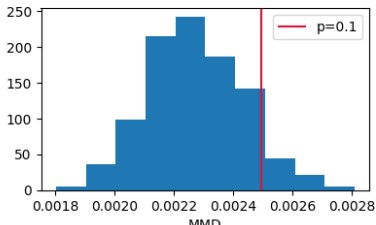 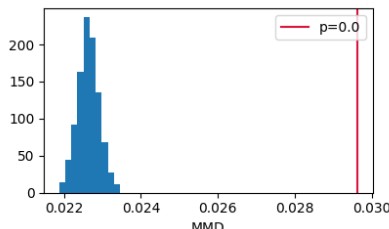

Figure 7: *Left*: Distribution of $\text{MMD}_u$ under Hypothesis $\mathcal{P}(\text{RNAStr}_{\text{train}}) = \mathcal{P}(\text{RNAStr}_{\text{test}})$. *Right*: Distribution of $\text{MMD}_u$ under Hypothesis $\mathcal{P}(\text{RNAStr}_{\text{train}}) = \mathcal{P}(\text{ArchiveII})$.

The result shows

(a) Hypothesis $\mathcal{P}(\text{RNAStr}_{\text{train}}) = \mathcal{P}(\text{RNAStr}_{\text{test}})$ can be accepted with significance level 0.1.

(b) Hypothesis $\mathcal{P}(\text{RNAStr}_{\text{train}}) = \mathcal{P}(\text{ArchiveII})$ is rejected since the p-value is 0.

Therefore, the data distribution in ArchiveII is very different from the RNAStralign training set. A good performance on ArchiveII shows a significant generalization power of E2Efold.

### D.3 PERFORMANCE ON LONG SEQUENCES: WEIGHTED F1 SCORE

For long sequences, E2Efold still performs better than other methods. We compute F1 scores weighted by the length of sequences (Table 8), such that the results are more dominated by longer sequences.

Table 8: RNAStralign: F1 after a weighted average by sequence length.

| Method | E2Efold | CDPfold | LinearFold | Mfold | RNAstructure | RNAfold | CONTRAfold |
|---|---|---|---|---|---|---|---|
| non-weighted | 0.821 | 0.614 | 0.609 | 0.420 | 0.550 | 0.540 | 0.633 |
| weighted | 0.720 | 0.691 | 0.509 | 0.366 | 0.471 | 0.444 | 0.542 |
| change | -12.3% | +12.5% | -16.4% | -12.8% | -14.3% | -17.7% | -14.3% |

The third row reports how much F1 score drops after reweighting.

### D.4 ARCHIVEII RESULTS AFTER DOMAIN SEQUENCES ARE REMOVED

Since domain sequence (subsequences) in ArchiveII are explicitly labeled, we filter them out in ArchiveII and recompute the F1 scores (Table 9).

The results do not change too much before or after filtering out subsequences.

Table 9: ArchiveII: F1 after subsequences are filtered out.

| Method | E2Efold | CDPfold | LinearFold | Mfold | RNAstructure | RNAfold | CONTRAfold |
|---|---|---|---|---|---|---|---|
| original | 0.704 | 0.597 | 0.647 | 0.421 | 0.613 | 0.615 | 0.662 |
| filtered | 0.723 | 0.605 | 0.645 | 0.419 | 0.611 | 0.615 | 0.659 |

## D.5 PER-FAMILY PERFORMANCES

To balance the performance among different families, during the training phase we conducted weighted sampling of the data based on their family size. With weighted sampling, the overall F1 score (S) is 0.83, which is the same as when we did equal-weighted sampling. The per-family results are shown in Table 10.

Table 10: RNAStralign: per-family performances

| | 16S rRNA | | tRNA | | 5S RNA | | SRP | |
|---|---|---|---|---|---|---|---|---|
| | F1 | F1(S) | F1 | F1(S) | F1 | F1(S) | F1 | F1(S) |
| E2Efold | 0.783 | 0.795 | 0.917 | 0.939 | 0.906 | 0.936 | 0.550 | 0.614 |
| LinearFold | 0.493 | 0.504 | 0.734 | 0.739 | 0.713 | 0.738 | 0.618 | 0.648 |
| Mfold | 0.362 | 0.373 | 0.662 | 0.675 | 0.356 | 0.367 | 0.350 | 0.378 |
| RNAstructure | 0.464 | 0.485 | 0.709 | 0.736 | 0.578 | 0.597 | 0.579 | 0.617 |
| RNAfold | 0.430 | 0.449 | 0.695 | 0.706 | 0.592 | 0.612 | 0.617 | 0.651 |
| CONTRAfold | 0.529 | 0.546 | 0.758 | 0.765 | 0.717 | 0.740 | 0.563 | 0.596 |
| | tmRNA | | Group I intron | | RNaseP | | telomerase | |
| | F1 | F1(S) | F1 | F1(S) | F1 | F1(S) | F1 | F1(S) |
| E2Efold | 0.588 | 0.653 | 0.387 | 0.428 | 0.565 | 0.604 | 0.954 | 0.961 |
| LinearFold | 0.393 | 0.412 | 0.565 | 0.579 | 0.567 | 0.578 | 0.515 | 0.531 |
| Mfold | 0.290 | 0.308 | 0.483 | 0.498 | 0.562 | 0.579 | 0.403 | 0.531 |
| RNAstructure | 0.400 | 0.423 | 0.566 | 0.599 | 0.589 | 0.616 | 0.512 | 0.545 |
| RNAfold | 0.411 | 0.430 | 0.589 | 0.599 | 0.544 | 0.563 | 0.471 | 0.496 |
| CONTRAfold | 0.463 | 0.482 | 0.603 | 0.620 | 0.645 | 0.662 | 0.529 | 0.548 |

### D.6 MORE VISUALIZATION RESULTS

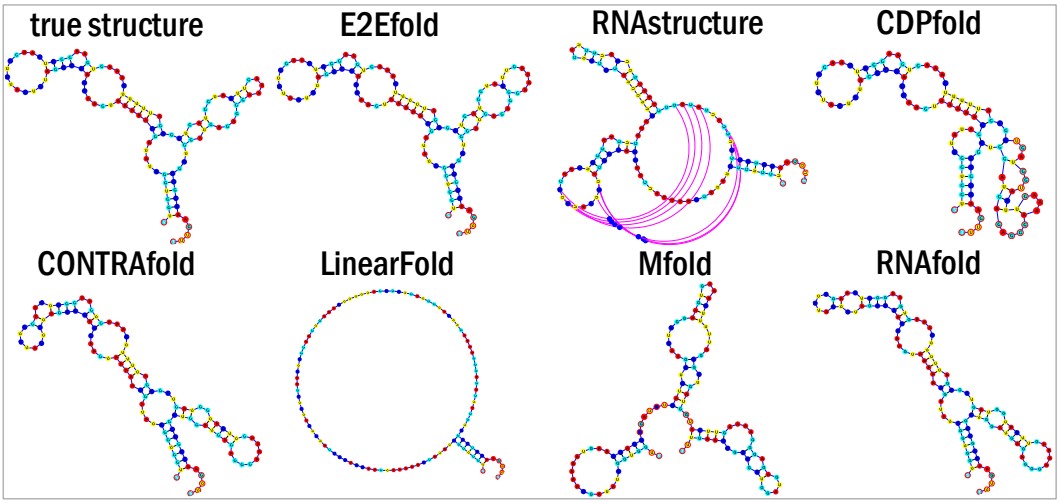

Figure 8: Visualization of 5S rRNA, B01865.

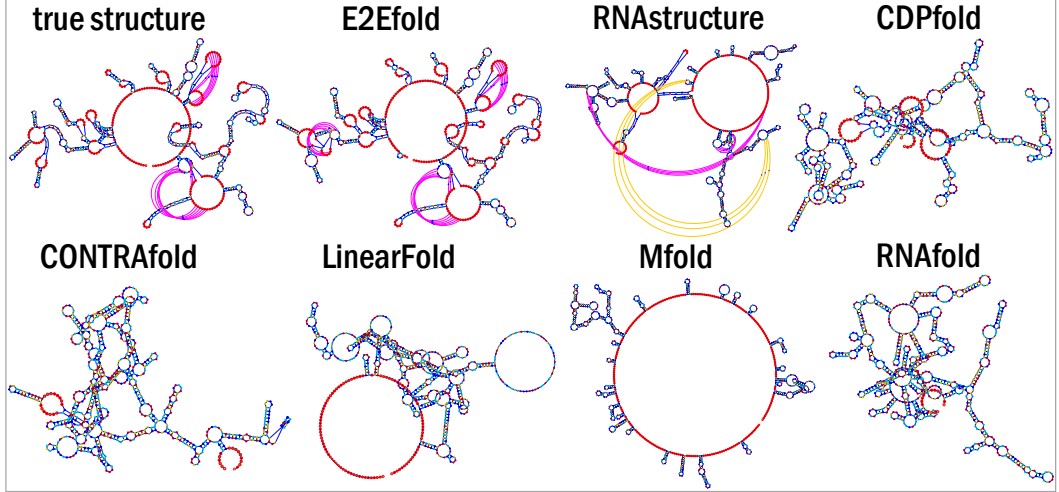

Figure 9: Visualization of 16S rRNA, DQ170870.

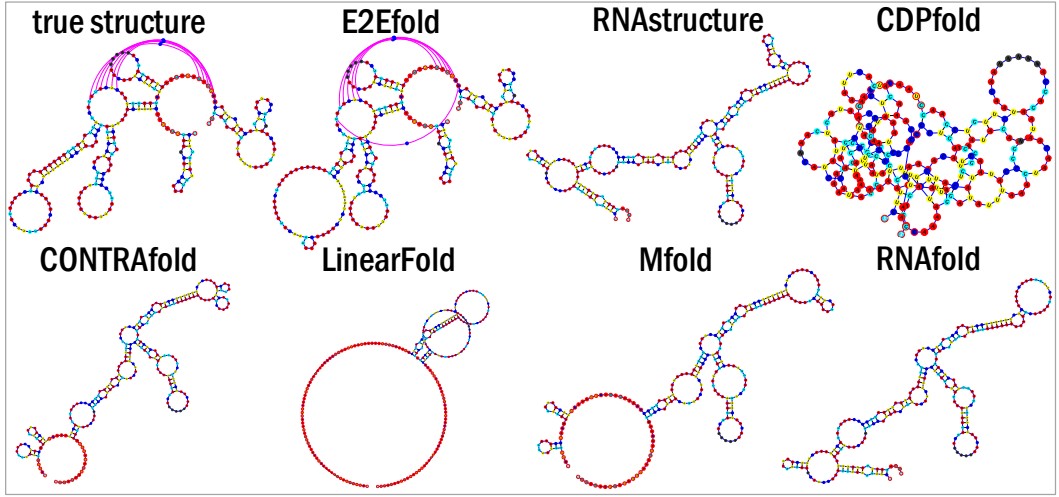

Figure 10: Visualization of Group I intron, IC3, Kaf.c.trnL.

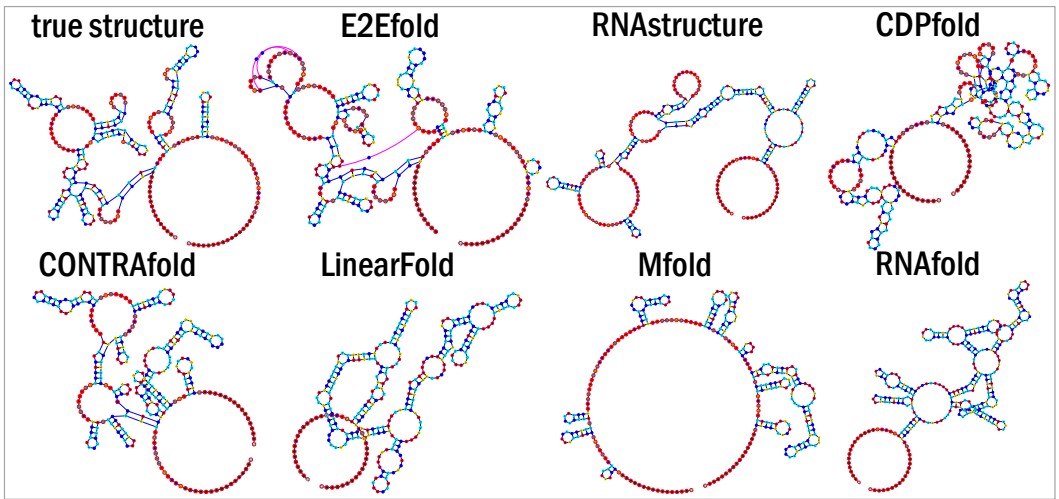

Figure 11: Visualization of RNaseP, A.salinestris-184.

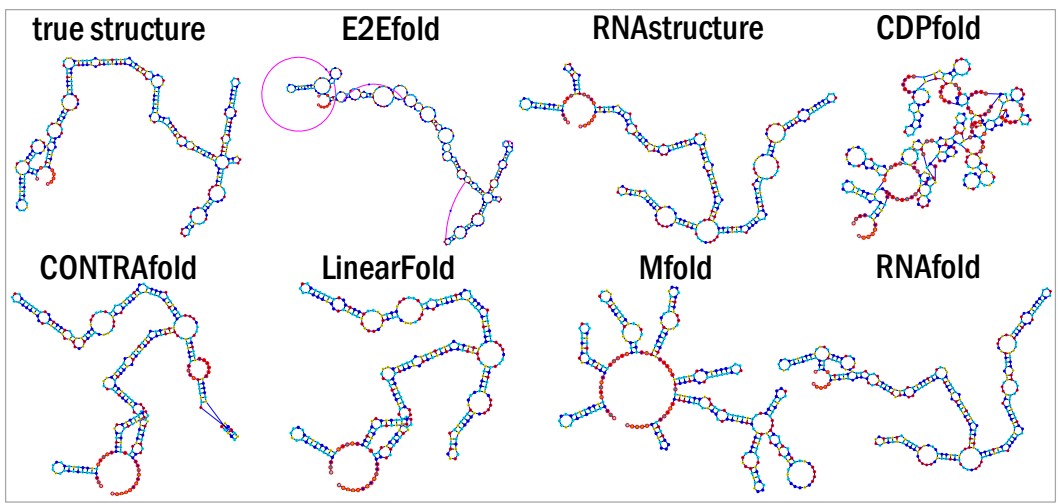

Figure 12: Visualization of SRP, Homo.sapi._BU56690.

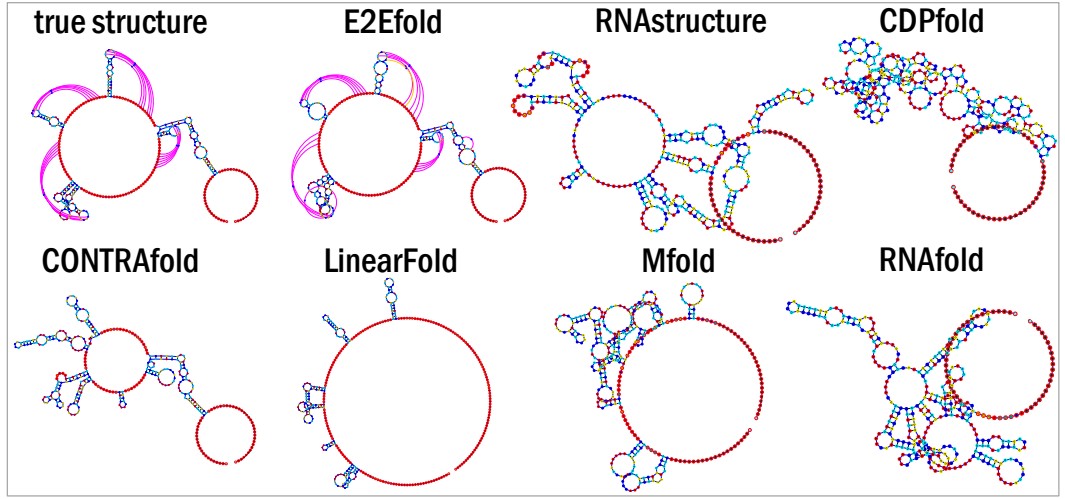

Figure 13: Visualization of tmRNA, uncu.bact._AF389956.

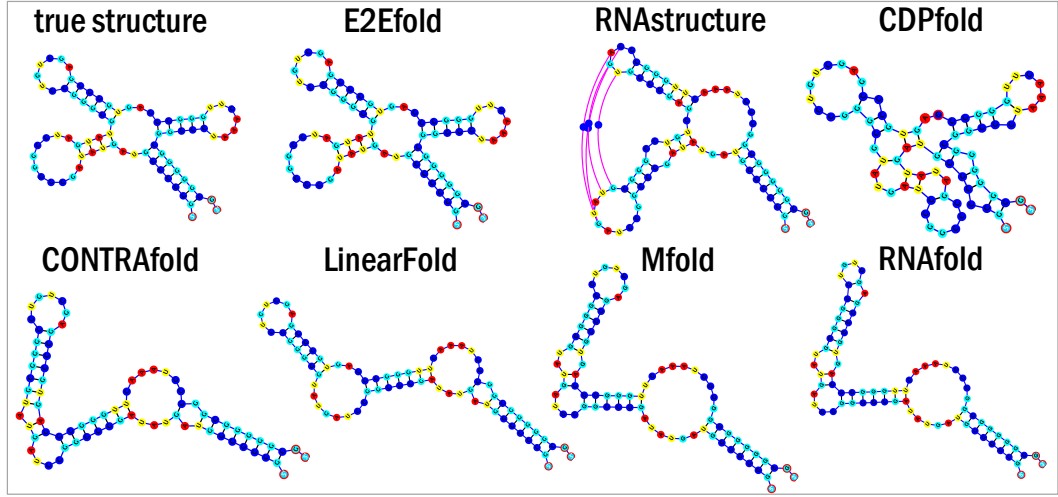

Figure 14: Visualization of tRNA, tdbD00012019.

