# OpenReview forum: "RNA Secondary Structure Prediction By Learning Unrolled Algorithms"
_ICLR.cc/2020/Conference — Accept (Talk)_

### Official Review · AnonReviewer3 · 2019-10-24
**Official Blind Review #3**

**Rating:** 6

**Review:**

*Summary*
The authors perform RNA secondary prediction using deep learning. The outputs are subject to hard constraints on which nucleotides can be in contact with others. They unroll a sophisticated optimization algorithm for a relaxation of the task of finding the optimal contact map subject to these constraints. This work is in a long line of work demonstrating that end-to-end training of models that incorporate application-specific optimization routines as sub-modules is very useful. In particular, it outperforms an approach where the inputs to this optimization problem come from a network that was trained using a simple loss that ignores the fact that it will feed into this structured optimizer. The paper also considers an application domain that will be unfamiliar to many ICLR readers interested in deep structured prediction, and may serve as a call to arms for the community engaging with additional problems in this field.

*Overall Assessment*
The paper is well written, well executed, and part of a general research thread that ICLR readers care about. There are a number of technical details, such as the loss function in (8) that will be of general interest. I advocate for acceptance.

*Comments*
The actual specification of the output constraints doesn't occur until late in the paper. Before then, the discussion of them is very abstract. Given that the constraints are easy to describe, the exposition would be improved notably if you described the specific constraints earlier on. This would help me understand the problem domain better.

Fyi, the idea of nested structures vs. non-nested structures appears in NLP in terms of projective parsing vs. non-projective parsing. There may be some relevant reading for you to do there. Your specific work (minus the unrolled constraint enforcement) is similar to "Dozat et al. 2017. Deep biaffine attention for neural dependency parsing."

The idea of backpropping through some constraint-enforcing process is reminiscent of backpropping through belief propagation. See, for example, Domke's "Learning Graphical Model Parameters with Approximate Marginals Inference." Or Hershey et al. "Deep Unfolding: Model-Based Inspiration of Novel Deep Architectures." You should also cite work using unrolled ISTA to learn sparse coding dictionaries. They have terms similar to (5).

What exactly was your motivation for the setup in "Test On ArchiveII Without Re-training?"

How sensitive is performance to the number of optimizer iterations? Does it work to train with a fixed number of unrolled iters, but at test time run the optimizer until convergence?

(8) is cool!


**Experience Assessment:**

I have published in this field for several years.

**Review Assessment: Checking Correctness Of Derivations And Theory:**

N/A

**Review Assessment: Checking Correctness Of Experiments:**

I assessed the sensibility of the experiments.

**Review Assessment: Thoroughness In Paper Reading:**

I read the paper at least twice and used my best judgement in assessing the paper.

---

> ### Author Response · Authors · 2019-11-12
> **Response To Official Blind Review #3**
>
> We would like to thank the reviewer for the overall positive comments, constructive suggestions on paper refinement and references to interesting related works!
>
>
> ***Q1.  “...output constraints doesn't occur until late...”
>
> Thank you for your suggestion on the flow of our paper! In the revised version, we state the RNA secondary structure prediction problem, including the concrete constraints, in a newly added section “3 RNA Secondary Structure Prediction Problem”, which follows the Related Work section.
>
>
> ***Q2. Related work in NLP
>
> Thank you for referring us to these related works in NLP! We also noticed the relation to NLP as we mentioned at the end of the introduction section. This indeed motivates us to use transformers and also the trick mentioned in BERT to compute position information by a series of base functions. However, it is interesting to know about projective parsing vs. non-projective parsing which we didn’t notice before! We’ve added a paragraph in the related work section to discuss this aspect. Thank you for pointing it out, which can help us relate our work to a larger range of problems in ML.
>
>
> ***Q3. Related work on Graphical Models and ISTA
>
> Thanks! We found the “deep unfolding” work very related and cited it in the revised paper.
> In our first submission, we’ve cited the unrolled ISTA paper “Theoretical linear convergence of unfolded ISTA and its practical weights and thresholds.”
>
>
> ***Q4. Motivation for the setup in "Test On ArchiveII Without Re-training"
>
> One can think of ArchiveII as a separate held-out test dataset. E2Efold is only learned from RNAStralign training set, but can directly generalize to ArchiveII, and obtain the best test results.
>
> In fact, testing on the ArchiveII dataset is a more challenging test for generalization, because the data distribution in ArchiveII can have a larger difference with the RNAStralign training set. To see this, we performed additional statistical hypothesis tests using Maximum Mean Discrepancy (MMD)  [1] and attached the results in Appendix D.2.
>
> More specifically, we computed the empirical MMD to evaluate the differences between
> (a) RNAStralign_train and RNAStralign_test, where the MMD is 0.0025 (*can not reject* null hypothesis of no difference with p-value 0.1)
> (b) RNAStralign_train and ArchiveII, where the MMD is 0.0296 (*reject* null hypothesis of no difference with p-value < 0.001)
> These tests suggest that the difference between RNAStralign training set and ArchiveII is much larger.
>
> Therefore, the data distribution in ArchiveII is very different from the RNAStralign training set. A good performance on ArchiveII shows a significant generalization power of our model.
>
>
> ***Q5. the number of optimizer iterations
>
> For the explanation on the choice/effect of the number of optimizer iterations, please kindly refer to our response to this common question posted above.
>
>
> ***Q6. Does it work to train with a fixed number of unrolled iters, but at test time run the optimizer until convergence?”
>
> It’s very interesting that the reviewer asked this question! We were also curious about this before and tried it empirically. We trained the model on T=20 and used it for T=50 iterations during the test phase. However, the performance is not as good as keeping T=20 for the test.
>
> We think the reason could be: we choose the discounting factor $\gamma=1$ in equation 9, which is also a default choice in some related works. It gives the output at each time step T=t an equal weight. With a smaller $\gamma$, the outputs at later steps can gain more weights. In this case, it is possible that the trained network will output a progressively closer approximation of the ground truth and further generalize to a larger number of iterations. We would investigate this option in the future.
>
>
> ***Finally
>
> Yes, we also like the differentiable F1 score! Imbalanced data (more negative samples than positive samples) is a common issue in many computational biology problems (e.g. [4,5]) and our proposed method is very simple and effective in this case.
>
>
> [1] Gretton, Arthur, et al. "A kernel two-sample test." JMLR (2012)
> [4] Zhou, Jian, and Olga G. Troyanskaya. "Predicting effects of noncoding variants with deep learning–based sequence model." Nature methods 12.10 (2015): 931.
> [5] Armenteros, José Juan Almagro, et al. "SignalP 5.0 improves signal peptide predictions using deep neural networks." Nature biotechnology 37.4 (2019): 420.
>
>
> Please expect the revised paper posted soon.

---

### Official Review · AnonReviewer1 · 2019-10-24
**Official Blind Review #1**

**Rating:** 8

**Review:**

The authors proposed an end-to-end method (E2Efold) to predict RNA secondary structure. The method consists of a Deep Score Network and a Post-Process Network (PPN). The two networks are trained jointly. The score network is a deep learning model with transformer and convolution layers, and the post-process network is solving a constrained optimization problem with an T-step unrolled algorithm. Experimental results demonstrate that the proposed approach outperforms other RNA secondary structure estimation approaches.

Overall I found the paper interesting. Although the writing can be improved and some important details are missing.

Major comments
As the authors point out, several existing approaches for unrolling optimization problems have been proposed. It would be helpful to clarify the methodological novelty of the proposed algorithm compared to those.


Training details and implementation details are missing; these hinder the reproducibility of the proposed approach. The author stated pre-training of the score network, how is the PPN and score network updated during the joint training? Does the model always converge? The authors vaguely mentioned add additional logistic regression loss to Eq9 for regularization. What is a typical number of T? How does varying T affect the performance, both in terms of training time (and convergence) and in terms of accuracy/F1?

Minor comments
The 29.7% improvement of F1 score overstates the improvements compared to non-learning approaches.. This performance was computed on the dataset (RNAStralign) on which E2Efold was trained. A fair comparison, as the authors also stated, is on the independent ArchiveII data. On this data, E2Efold has F1 score 0.686 versus 0.638 for CONTRAfold. The author should report performance improvement under this line.


It would be helpful to report performance per RNA category, both for RNAstralign data and ArchiveII data, while the ArchiveII data should still remain independent. Different models may have their strengths and weaknesses on different RNA types.


It is not obvious to me how the proximal gradient was derived to (3)-(5). It would be helpful if the authors show some details in the supplements.


Why is there a need to introduce an l_1 penalty term to make A sparse?


On which data is Table 6?

Typos, etc.
The references are not consistently formatted
“structure a result” -> “structure is a result”
“a few hundred.” -> “a few hundred base pairs.”
“objective measure the” -> “objective measures the”
“section 5” -> “Section 5” (in several places)
In the equation above Equation 2, should it be -\rho||\hat{A}||_{1} instead of plus? Otherwise, the “max” could be made arbitrarily large.


**Experience Assessment:**

I do not know much about this area.

**Review Assessment: Checking Correctness Of Derivations And Theory:**

I did not assess the derivations or theory.

**Review Assessment: Checking Correctness Of Experiments:**

I assessed the sensibility of the experiments.

**Review Assessment: Thoroughness In Paper Reading:**

I read the paper at least twice and used my best judgement in assessing the paper.

---

> ### Author Response · Authors · 2019-11-12
> **Response To Official Blind Review #1 (part 1)**
>
> We thank reviewer 1 for careful reading and constructive suggestions for paper refinement! We separate our response to reviewer 1 into two parts.
>
> The first part of reviewer 1’s comments is about missed important details. We are sorry for the lack of clarity and thank the reviewer for pointing them out! Now the details are included in the revised paper, and we also explained them below:
>
>
> ***Compare to existing approaches for unrolling optimization problems
>
> We explained the novelty/difference of E2Efold compared to the existing approaches below. They are now included in the main text and appendix A.
>
> First, our view of incorporating constraints to reduce output space and to reduce sample complexity is novel. Previous works [Belanger et al., 2017; Pillutla et al., 2018; Ingraham et al., 2018]  did not discuss these aspects. The most related work which also integrates constraints is OptNet [Amos & Kolter, 2017], but it’s very expensive and can not scale to the RNA problem. Therefore, our proposed approach is a simple and effective one.
>
> Second, compared to [Andrychowicz et al., 2016, Chen et al., 2018; Shrivastava et al., 2019], our approach has a different purpose of using the unrolled algorithm. Their goal is to learn a better algorithm, so they commonly make the architecture *more flexible* than the original algorithm for the room of improvement. However, we aim at enforcing constraints. To ensure that constraints are nicely incorporated, we keep the original structure of the algorithm and only make the hyperparameters learnable.
>
> Finally, although all works consider end-to-end training, none of them can directly optimize the F1 score. We proposed a differentiable loss function to mimic the F1 score/precision/recall, which is effective and also very useful when negative samples are much fewer than positive samples (or the inverse).
>
>
> ***Details on the unrolling constant T
>
> For the explanation on the choice/effect of unrolling constant T, please kindly refer to our response to this common question posted above.
>
>
> ***Details on the design of the training process
>
> Now all the training details are included in Appendix C of the revised paper.
>
> Connecting the score network and PPN, the whole network is very deep. Both the pre-training and the augmented loss are the tricks that we use to speed up and stabilize the training process.
> - Pre-train: Since we expect $U_{\theta}(i,j)$ to be higher if $(x_i,x_j)$ are truly paired, we use the true secondary structure to pre-train the score network to quickly get a fairly good U.
> - Augmented loss: During joint training, we use (loss in Eq 9) + $c \cdot$(logistic regression loss on $U_{\theta}$) where we set $c=1$. The second term is estimated at the *intermediate output*. Although this term is optional, we think it can help stabilize the training of the *very deep* network and also make each gradient step more efficient.
>
> Besides, we’ve submitted a link to our code through a private comment to reviewers. We will get this code well-organized and released to the public for others to reproduce all the experimental results.

---

> > ### Author Response · Authors · 2019-11-12
> > **Response To Official Blind Review #1 (part 2)**
> >
> > Here we provided our responses to the second part of reviewer 1’s comments.
> >
> > ***Overstates the improvements
> >
> > According to the reviewer’s suggestion, we’ve moderated our statement by simply saying “better than previous SOTA in terms of F1 scores”.
> > Btw, the performances on RNAStralign are estimated on a held-out test set, so we think the comparison on this test set is also valuable.
> >
> >
> > ***performance per RNA category
> >
> > To balance the performance among different families, during the training phase we conducted weighted sampling of the data based on their family size.  With weighted sampling, the overall F1 score (with shifted) is 0.83, which is the same as when we did i.i.d. sampling. The per-family F1 scores are provided below. More numbers are included in Appendix D.5 in the revised paper.
> >
> > Table 1: RNAStralign (test set): per-family F1 score
> >                          |16S rRNA| tRNA | 5S RNA |  SRP | tmRNA |  Grp I  | RNaseP | telomerase
> > family size      |    11620  |  9385 |   6443   |  468   |    572   |  1502   |     434    |   37
> > E2Efold           |    0.783   | 0.917 |   0.906   | 0.550 |  0.588  | 0.387  |   0.565   |  0.954
> > LinearFold     |   0.493    | 0.734 |   0.713   | 0.618 |  0.393  | 0.565  |   0.567   |  0.515
> > Mfold              |    0.362   | 0.662 |   0.356   | 0.350 |  0.290  | 0.483  |   0.562   |  0.403
> > RNAstructure|    0.464   | 0.709 |   0.578   | 0.579 |  0.400  | 0.566  |   0.589   |  0.512
> > RNAfold          |    0.430   | 0.695 |   0.592   | 0.617 |  0.411  | 0.589  |   0.544   |  0.471
> > CONTRAfold  |    0.529   |  0.758 |   0.717  | 0.563 |  0.463  |  0.603 |   0.645   |  0.529
> >
> > E2Efold performs significantly better than other methods in 16S rRNA, tRNA, 5S RNA, tmRNA, and telomerase, and these results are from *a single model*.
> >
> > In the future, we can view it as multi-task learning and further improve the performance by learning multiple models for different families and learning an additional classifier to predict which model to use for the input sequence.
> >
> >
> > ***Q3. Derivation for the proximal gradient steps
> >
> > We added include the derivation steps for Eq. 3-5 in Appendix B in the revised version.
> >
> >
> > ***Q4. Why is there a need to introduce an l_1 penalty term to make A sparse?
> >
> > This is mainly due to the *fixed* number of iterations T. We prefer using a comparatively *small T* (e.g. T=20) as we explained above. In this case, we use a sparse penalty to help the algorithm quickly incorporate the constraints to the output within T iterations, since it is our prior knowledge that the global constraints (e.g. each $x_i$ can only be paired with at most *one* base $x_j$) will give us a sparse output.
> >
> > For example, within T=20 iterations, we compare the differences between adding and not adding the sparse penalty:
> >
> > Table 2: Constraints-check.
> >            		                   | $\max_{A}c(A)$ | $\text{mean}_{A}c(A)$
> > With sparse penalty       |           4         |     0.65
> > Without sparse penalty | 	       28        |    3.975
> >
> > In Table 1, $c(A):=\text{sum}(\text{relu}(A\mathbf{1} - \mathbf{1}))$ is measuring whether the constraints are satisfied. The smaller the better. The benefit of adding the penalty term is obvious, in the case when T is fixed and comparatively small.
> >
> >
> > ***Q5. On which data is Table 6?
> >
> > RNAStralign. We are sorry for the lack of clarity. We’ve indicated the dataset in the caption of Table 6 in the revised version.
> >
> >
> > ***grammar error/typo
> >
> > Yes! In the equation above Equation 2, it should be $-\rho||\hat{A}||_1$ instead of plus! Thank you for pointing all the typos/errors out! We believe they are now fixed.
> >
> >
> > Please expect the revised paper posted soon.

---

> > > ### Comment · AnonReviewer1 · 2019-11-15
> > > **RE: Author response**
> > >
> > > I have read the other reviews (and comments from Liang and Thomas). My view of the paper ("Accept") is unchanged after reading that discussion.
> > >
> > > I believe it would be helpful to include a bit of that discussion in the paper (especially about how energy functions are commonly learned from data as well, just in an "offline", non-differentiable step), and certainly to include the additional results in a table or appendix, etc. I completely agree that it is important to make clear that the training and testing sets are different.
> > >
> > > Further, even if we accept the criticisms of Liang and Thomas on the performance evaluation (again, I believe the authors credibly addressed these criticisms), there is still nice methodological contribution in this paper. As AnonReviewer3 said: "(8) is cool!"

---

> > > > ### Author Response · Authors · 2019-11-15
> > > > **Thank you very much for your positive feedback!**
> > > >
> > > > Thank you very much for your positive feedback! Yes, we agree with you and we will keep refining the manuscript to make the final version more comprehensive.

---

### Official Review · AnonReviewer2 · 2019-10-28
**Official Blind Review #2**

**Rating:** 8

**Review:**

This paper introduces an end-to-end method to predict the secondary structure of RNA, by mapping the nucleotide sequence to a binary affinity matrix. The authors decompose this problem into two part: (i) predicting an affinity score between each base pair in the input sequence, using a combination of a transformer sequence encoder network and a convolutional decoder, and (ii) a post-processing step that ensures that structural local and global constraints are enforced. An innovation is to express this post-processing as an unrolled sequence of proximal gradient descent steps, which are fully differentiable, and allow the full combination of (i)+(ii) to be trained end-to-end. A thorough set of experiments validate the approach.

Overall, the paper is well written and easy to follow. The approach of unrolling structural constraints as shown in the paper is interesting and applicable to much wider domains than secondary structure prediction. The proposed approach appears to provide a novel, convincing and non-obvious solution to RNA secondary structure prediction, and subject to suggestions below, would represent a valuable contribution to ICLR.

The principal area for improvement would be to include additional detail (perhaps in appendix) on the model hyperparameter configurations that were used in the experiments. Moreover, more details on the set of \psi functions, and the MLP details for P_i (e.g. number of hidden units, activation function, the use of dropout, batch normalization, etc) should be given, as well as more information on the specifics how how the “pairwise concatenation” is carried out in the output layer. What unrolling constant T is used? Finally, in the ablation study (p. 8) details on how U_\theta is trained by itself (without the post-processing step) should be given.

Detailed comments:
* Overall, the whole paper should be thoroughly reviewed for English grammar and writing style; a subset of suggested changes follow.
* p. 1: structure a result ==> structure is a result
* p. 2: energy based methods ==> energy-based methods
* p. 2: energy function based approaches ==> energy function-based approaches
* p. 2: view point ==> viewpoint
* p. 2: E2Efold is flexible ==> E2Efold are flexible
* p. 2: nearly efficient ==> nearly efficiently
* p. 3: typically scale ==> typically scale as
* p. 3: few hundred. ==> few hundreds.
* p. 4: all binary matrix ==> all binary matrices
* p. 4: output space can help ==> output space could help
* p. 5: formulation are the ==> formulation are that the
* p. 6: eq. (7) should contain quantities indexed by $t$ in the RHS
* p. 8: pesudoknotted ==> pseudoknotted
* p. 9 ff: in the bibliography, all lowercase rna should be uppercase RNA. Use {RNA} in bibtex entries.


**Experience Assessment:**

I do not know much about this area.

**Review Assessment: Checking Correctness Of Derivations And Theory:**

N/A

**Review Assessment: Checking Correctness Of Experiments:**

I assessed the sensibility of the experiments.

**Review Assessment: Thoroughness In Paper Reading:**

I read the paper at least twice and used my best judgement in assessing the paper.

---

> ### Author Response · Authors · 2019-11-12
> **Response To Official Blind Review #2**
>
>
> We thank the reviewer for the positive comments about the approach and the suggestions for paper refinement! We’ve included all the important details suggested by the reviewer in either the main paper or appendix. Experimental results are demonstrated to show how we choose the unrolling constant T.
>
>
> ***Q1. details of the configuration
>
> We’ve added a section (Appendix C) to explain our configurations including hyperparameters, MLP details, and the ‘pairwise concatenation’ details.
>
> Besides, we’ve submitted a link to our code through a private comment to reviewers. We will get this code well-organized and released to the public for others to reproduce all the experimental results.
>
>
> ***Q2. “What unrolling constant T is used? ”
>
> For the explanation on the choice/effect of unrolling constant T, please kindly refer to our response to this common question posted above.
>
>
> ***Q3. details on how U_\theta is trained by itself
>
> We are sorry for the lack of clarity in our first submission. Now all the training details are included in the revised paper. Please kindly refer to Appendix C. Briefly speaking, the training process of $U_{\theta}$ is the same as E2Efold, which consists of two steps. First, quickly pre-train $U_{\theta}$ using logistic regression. Second, use the loss in eq 9 to jointly train $U_{\theta}$ and $PP_{\phi}$ end-2-end (when $U_{\theta}$ is trained by itself we can simply set T=0 in the PP network).
>
>
> ***Q4. English grammar and writing style
> We appreciate the reviewer for the very close reading! We believe the typos/errors are now fixed.
>
>
> Please expect the revised paper posted soon.

---

### Official Review · AnonReviewer4 · 2019-11-22
**Official Blind Review #4**

**Rating:** 8

**Review:**

RNA Secondary Structure Prediction by Learning Unrolled Algorithms

This paper proposes E2Efold, which is an RNA secondary structure prediction algorithm based on an unrolled algorithm. Previous methods rely on dynamic programming (which does not work for molecular configurations that do not factorize) or rely on energy-based models (which require a minimization step, e.g. by using MCMC to traverse the energy landscape and find minima). The former does not work for all molecules and the latter can be difficult to optimize. The method presented here is novel, shows strong SOTA performance, and would be of interest to the wider deep learning community.

The method is based on an unrolled algorithm, which is motivated by the inclusion of three inductive biases / constraints important underlying RNA folding. These constraints limit the wide RNA search space. The first component of the method is a “Deep Score Network” which uses a stack of Transformer encoders (with relative and exact positional embeddings) followed by 2D convolutional layers to output a L x L symmetric matrix describing the “scores” of base pairing. As these scores may not obey the rules of RNA folding, a second post-processing network is trained end-to-end together with the “Deep Score Network” to enforce constraints. This network starts with a transformation that symmetrizes the matrix and applies a constraint-enforcing mask. The problem is transformed into an unconstrained problem by using Lagrange multipliers; it is then solved using a proximal gradient. Finally, a recurrent cell is defined that implements this algorithm in a deep learning framework. This method is creative, could be applied to other tasks with constraints, and would be interesting to the wider deep learning community.

In addition to developing the deep score network and post-processing network, the authors also develop a differentiable F1 loss, so that the network can directly optimize for precision and recall on the task. The performance of this method significantly outperforms previous methods. There was a fruitful discussion on OpenReview regarding whether this was a result of overfitting on the task. Indeed, it is critical in deep learning applications to carefully construct train/test sets to avoid high performance by memorization alone. To address this, the authors train on RNAStralign and test on ArchiveII. As the original ArchiveII dataset contains subsequences of other RNA sequences, which can result in overfitting, the authors re-ran their experiment with that removed, and similar results were achieved. To support the hypothesis that ArchiveII and RNAStralign capture different distributions, they perform a permutation test on the unbiased empirical Maximum Mean Discrepancy estimator, finding that the distributions are different. I do wonder why they did not check if P(ArchiveII) = P(ArchiveII) as they do check if P(RNAStr_train) = P(RNAStr_train). On the specific task of pseudoknot prediction, the method also performs well (F1 is >0.23 over the baseline). On sequence length-weighted F1, the model does even better.

The paper is rich with ablations. The analysis of the number of unrolling iterations T helps support the use of an unrolled method and builds intuition for its importance - it would be useful to include this in the appendix of the paper. I also appreciated the visualizations, which are a good sanity check that the model correctly handles pseudoknots. The performance of the method is broken down by RNA family, which is also quite interesting -- the method outperforms LinearFold on all classes, besides 5S RNA, SRP, and Group I intron. Further analysis is required to better understand why the method is weaker on those datasets. Additionally, further work should explore training on one set of families and testing on a held-out set of families. This was pointed out by public comments on this paper. This is potentially a limitation of E2EFold (the authors do not seem to have tried this suggested experiment) and further exploration is required. Exploring this limitation (even if it is not overcome) would make this paper even more rich.

That said, I recommend acceptance of this paper due to the extensive experiments, polished writing, novel method, and strong results, which can inspire future research.


**Experience Assessment:**

I have read many papers in this area.

**Review Assessment: Checking Correctness Of Derivations And Theory:**

I assessed the sensibility of the derivations and theory.

**Review Assessment: Checking Correctness Of Experiments:**

I carefully checked the experiments.

**Review Assessment: Thoroughness In Paper Reading:**

I read the paper thoroughly.

---

### Public Comment · ~Liang_Huang1 · 2019-11-01
**interesting idea but seriously flawed in evaluation**

As the primary author of the LinearFold paper (Huang et al, 2019) which this submission compares to, I found this deep learning approach novel and interesting, being vastly different from the mainstream dynamic programming (DP)-based algorithms. However, the evaluation in this work is seriously flawed in many ways:

1. Dataset: (a) mistakenly using subsequences as full sequences

This paper uses two datasets from the Mathews group: RNAStralign and ArchiveII. However, the authors did not notice that many sequences in the 16S and 23S families are labeled as *sub domains* and thus sub sequences, not full sequences. For example, 16S is family of ~1,500 nts with length in the range of [950,1995] in ArchiveII, while in Table 1 of this paper, 16S's minimum length is only 73 and 54 for the two datasets. This obvious mistake shortens the average length of the datasets considerably.

2. Dataset: (b) each family is extremely homogeneous in sequences and structures

This paper is mainly trained and tested on RNAStralign, but that dataset was made for homologous folding -- meaning within each family (such as tRNA), all sequences are very similar to each other, and they share almost the same structures. As an example, let's see the first 5 sequences in tRNA:

1 72 tdbD00000002.ct
GGGCUCAUAGCUCAGCGGUAGAGUGCCUCCCUUGCAAGGAGGAUGCCCUGGGUUCGAAUCCCAGUGAGUCCA
(((((((..((((.......)))).(((((.......))))).....(((((.......)))))))))))).
2 72 tdbD00000003.ct
GGGCUCAUAGCUCAGCGGUAGAGUGCCUCCUUUGCAAGGAGGAUGCCCUGGGUUCGAAUCCCAGUGAGUCCA
(((((((..((((.......)))).(((((.......))))).....(((((.......)))))))))))).
3 72 tdbD00000004.ct
GGGCUCAUCGCUCAGCGGUAGAGUGCCUCCCUUGCAAGGAGGAUGCCCUGGGUUCGAAUCCCAGUGAGUCCA
(((((((..((((.......)))).(((((.......))))).....(((((.......)))))))))))).
4 72 tdbD00000005.ct
GGGCUCGUAGCUCAGCGGGAGAGCGCCGCCUUUGCGAGGCGGAGGCCGCGGGUUCAAAUCCCGCCGAGUCCA
(((((((..((((.......)))).(((((.......))))).....(((((.......)))))))))))).
5 72 tdbD00000006.ct
GGGCUCGUAGCUCAGCGGGAGAGCGCCGCCUUCGCGAGGCGGAGGCCGCGGGUUCAAAUCCCGCCGAGUCCA
(((((((..((((.......)))).(((((.......))))).....(((((.......)))))))))))).

Therefore, training and testing on the same families is always too easy and kind of overfitting. The correct way to do evaluation is the "cross-validation" way: training on N-1 families, and testing on the held out one, and repeat for all families and report the average.

3. Evaluation metrics

This paper only reports overall accuracy, but no per family accuracy. In addition, the overall accuracy is averaged over sequences, not families. By contrast, standard RNA folding evaluation should report per-family accuracies and while for each family, the accuracies are averaged over all sequences in this family, the overall accuracy should be averaged over all families (i.e., equal weight per family). Please refer to the LinearFold paper and other papers from the Mathews group which the authors cited (e.g., ProbKnot). In the authors' evaluation, the accuracy is dominated by short sequences (see Figure 6). They also admit prediction accuracy for some families are bad in Discussion: “For telomerase class, which only contains 37 samples in the dataset, E2Efold performs worse than classic methods.” Thus, their claim in abstract that “it predicts significantly better structures compared to previous SOTA (29.7% improvement in some cases in F1 scores and even larger improvement for pseudoknotted structures)” is misleading.

4. Model generalization:

As mentioned in 2, this paper does not apply cross-valid training and testing, instead they use RNAStralign-train (split from RNAStralign) as a training set, and RNAStralign-test and ArchiveII as testing sets. Moreover, they do not include all families in ArchiveII, but only the ones that have overlapping types (5S, 16S rRNA etc) in RNAStralign-train; i.e., they do not include 23S rRNA when testing on ArchiveII. Thus, their model’s generalization ability is suspicious and it may not work well on 23S rRNA. It is likely that their prediction examples perform well because of overfitting.

5. Runtime analysis

This paper claims they are faster than LinearFold and RNAstructure for inference time (see table 4). But they don’t provide the runtime with sequence length, instead they only compared the total runtime on RNAstralign. It is likely that their model costs much more time on long sequences. Moreover, the runtime is even tested on different hardware, for theirs on GPU, while LinearFold and RNAstructure on CPU. Besides, they do not report training time, which could be very costly.

6. (minor) RNAstructure vs Probknot

The authors seem to report Probknot as RNAstructure, but the latter (by default) can't predict pseudoknots.

Overall, this paper contains interesting ideas, but their evaluation is so flawed that their results are just overfitting. Cross-validation is absolutely needed for this type of data set where sequences/structures are so similar. Therefore, this paper can not be published in the current form.

---

> ### Author Response · Authors · 2019-11-03
> **Response: a further clarification (part 1)**
>
>
> We thank Liang Huang for your comments. We have carefully designed our experiments, and clarified further the settings below:
>
>
> *** Generalization ability
>
> ArchiveII is a separate held-out test dataset. E2Efold is only learned from RNAStralign training set, but can directly generalize to ArchiveII, and obtain the best test results.
>
> We note that LinearFold is not a learning-based method, but one energy function used in LinearFold is a learning-based energy from another paper (i.e. CONTRAfold), which can potentially be fitted from a much larger number of available datasets. Thus the per-family error of LinearFold is not a cross-validation error in the strict learning sense.
>
>
> *** No redundant sequences
>
> As stated in Sec 5, "After removing redundant sequences and structures...", we've carefully removed redundant sequences and structures, which reduces the RNAStralign dataset size from 37149 to 30451. Also, when we test the learned model on ArchiveII, we've excluded sequences that are overlapped with the RNAStralign dataset.
>
> The similarity between sequences is a characteristic of the dataset. Some level of similarity between sequences is also a basis for model generalization.
>
>
> *** Accuracy dominated by short-sequence?
>
> For long sequences, E2Efold still performs better than other methods. In fact, we've computed F1 scores *weighted by the length of sequences* (Table 1), such that the results are more dominated by longer sequences (we've conducted this experiment before the submission, but did not report them in the original submission due to page limit). For instance, for the RNAStralign dataset:
>
> Table 1. RNAStralign: F1 after a weighted average by sequence length.
>                  |E2Efold|CDPfold|LinearFold| Mfold |RNAstructure|RNAfold|CONTRAfold
> original   |  0.821  |  0.614   |    0.609     |  0.420  |       0.550       |   0.540  |     0.633
> weighted|  0.720  |  0.691   |    0.509     |  0.366  |       0.471       |   0.444  |     0.542
> change    |-12.3% | +12.5% |   -16.4%   | -12.8% |     -14.3%       | -17.7%  |   -14.3%
>
> 3rd row reports how much F1 score drops after reweighting.
>
>
> *** Subsequences
>
> Since subsequences in ArchiveII are explicitly labeled, we filtered them out in ArchiveII and recomputed the F1 scores (Table 2) as Liang suggested.
>
> Table 2. ArchiveII: F1 after subsequences are filtered out.
>               |E2Efold|CDPfold|LinearFold| Mfold |RNAstructure|RNAfold|CONTRAfold
> original| 0.704   |  0.597   |    0.647     |  0.421 |        0.613       |   0.615   |    0.662
> filtered | 0.723   |  0.605   |    0.645     |  0.419 |        0.611       |   0.615   |    0.659
>
> The results do not change too much before or after filtering out subsequences.
>
> In the original submission, we do not exclude subsequences by default because we want to follow RNA structure prediction benchmark. More specifically,
> - as suggested in the latest review by [Mathews, 2019], "The collection of benchmarking structures we collected, called ArchiveII, is available for download from our lab website at https://rna.urmc.rochester.edu/publications.html", ArchiveII is considered the benchmarking dataset and by default both domain sequences and full sequences are included.
> - the subsequences in ArchiveII are domains, whose secondary structure can be predicted independently since under most circumstances there is no cross-domain base-pairing.
>
> [Mathews, 2019] How to benchmark RNA secondary structure prediction accuracy, Methods, 2019
>
>
> *** Runtime comparison
>
> In practice, training needs to be done much less often than testing, and hence we focus on reporting the testing time. For instance, deep learning models for Imagenet need to be trained for a long time, but inference (testing) occurs more often, and lots of researches are done for improving inference time.
>
> Furthermore, it is undeniable that Dynamic Programming (DP) based algorithms are not easy to be parallelized or sped up by GPU due to their *sequential decision* nature. Thus, the runtime of DP-based algorithms has a significant dependency on the length n.
>
> In contrast, the operations in E2Efold are mainly *matrix computations*, which can be easily sped up by GPU using modern programming languages such as pytorch and tensorflow.
>
> Indeed, E2Efold will have even more advantages if we only report the runtime on long sequences. For example, we run E2Efold and LinearFold for sequences of length from 1500 to 1800 ("Long" in the table):
>           |E2Efold (Python)|LinearFold (C)|
>     All |           0.40s          |        0.43s        |
> Long|           0.41s          |        1.25s        |
>
> In addition, other methods are even implemented in C, which is by nature faster than Python that we used for E2Efold.

---

> > ### Author Response · Authors · 2019-11-03
> > **Response: a further clarification (part 2)**
> >
> >
> >
> > *** Probknot vs RNAstructure
> >
> > "The authors seem to report Probknot as RNAstructure, but the latter (by default) can't predict pseudoknots." : this statement is not true!
> >
> > RNAstructure web-server has been constantly updated, and Probknot is now included in the RNAstructure web-server:
> > https://rna.urmc.rochester.edu/RNAstructureWeb/
> > It does predict pseudoknots as shown in visualizations and table 5.
> >
> > *** Data requirement for DL-based methods
> >
> > The training data requirement cannot be avoided for DL models, so we do not deny the fact (in the discussion section) that E2Efold does not perform well on telomerase where there are only 37 data points. This problem needs to be resolved by future studies on few-shot learning or curriculum learning. Meanwhile, we expect that such training data will be accumulating in the future, too.
> >
> > For the past decades, DP-based algorithms have dominated the RNA secondary structure prediction. We believe that it is worth trying to apply deep learning to this problem even when some RNA families do not contain enough data at this moment. Furthermore, our methods present a more sensible bias-variance trade-off (deep while incorporating problem structure in the architecture), and thus a better result is reasonable.
> >
> > *** Methodology contribution
> >
> > Finally, we believe that the methodology proposed in our paper is of broad interest to the audience of the ML community. Though our work is originally motivated by the challenges in the RNA folding problem, we believe our ideas of
> > (i) incorporating hard constraints to reduce the output space and thus making it more data-efficient; and
> > (ii) using an unrolled algorithm for solving constrained programming as a building block in a neural network
> > will be inspiring for other structured prediction problems and potentially useful for architecture design in a wider range of DL problems. Therefore, we think that our work is well-suited for ICLR.

---

> > > ### Public Comment · ~Thomas_Litfin1 · 2019-11-05
> > > **Clarification of evaluation**
> > >
> > > I am in full agreement with the feedback of Liang Huang. The integration of hard constraints in the deep learning framework is a good idea which is well suited to the RNA structure prediction problem. However, I believe that the evaluation requires further clarification.
> > >
> > > RNA structure is remarkably conserved within a family even below 100% sequence identity. This means that we can consider 2 relevant settings for evaluation:
> > > 1) The setting whereby there exists known structures from the same family.
> > > 2) The setting whereby predictions are made for previously unseen families.
> > >
> > > The authors propose a method for setting 1 and compare to methods designed for setting 2. In addition, my understanding is that they evaluate competing methods considering only a single-sequence prediction. However, this may be an unfair comparison with E2Efold which is able to incorporate information from many family-specific homologs. Within the RNAstructure package, Turbofold may be used to generate family-specific consensus secondary structures from a collection of homologous sequences.
> > >
> > > They also briefly discuss a few-shot learning framework for families with few solved structures. In fact, a specific realisation of this framework is long-established in structural bioinformatics and is known in the field as homology modelling. The most naive version of this method is to align RNA sequences from the same family based on nucleotide substitution scores and transfer the known secondary structure to the unknown sequence based on the alignment. There are also more sophisticated realisations of this idea (eg Infernal). A comparison with a homology modelling method is necessary to evaluate the performance in setting 1, however the conservation of secondary structure within RNA families likely leaves little room for improvement over the alignment-based methods.
> > >
> > > However, the framework proposed by the authors can also be evaluated in setting 2 by evaluating the model on families unseen during training. Strong performance in this setting would certainly be important in the context of structural bioinformatics.

---

> > > > ### Author Response · Authors · 2019-11-11
> > > > **Results of the methods suggested by Thomas**
> > > >
> > > > We thank Thomas for referring us to more related methods. We’ve investigated and tried these methods (see more details below), although they are not commonly used as baselines for RNA folding.
> > > >
> > > > ***Turbofold is extremely slow and performs particularly bad on long sequences***
> > > >
> > > > We investigated the TurboFold package. It has serious flaws compared to our approach. TurboFold is extremely slow, requiring more than 15 minutes for only 5 sequences. This is true even for the parallel version, TurboFold-smp, for which we used 32 processors. Without multi-processing, the program won’t return any results for just 5 16S-rRNA inputs in a reasonable time. The running time of TurboFold seems to increase exponentially in the sequence length. Also, to run TurboFold for a set of sequences, the RNA class for each sequence must be known. Its performance, 0.75 F1, is worse than E2Efold. In addition, it performs particularly bad on *long* sequences, such as 16S rRNA (F1=0.5). E2Efold achieves 0.78 F1 in this class.
> > > >
> > > > The binary version of TurboFold has a specific input format requirement. We spent a very long time making the program successfully run on the whole testing dataset. Here is our code for running TurboFold:
> > > > https://drive.google.com/open?id=1HouOnpkN1vUf_NkZ3s6uKY0KL-c-YQD_
> > > >
> > > >
> > > > ***Infernal can not be used to predict RNA secondary structure***
> > > >
> > > > We’ve looked into the Infernal package. However, we found that (this is also pointed out in [1]):
> > > >
> > > > Infernal incorporates *the information of RNA secondary structure* (including that of the input sequence) to build a better homolog model than others for *searching similar sequences*.
> > > >
> > > > Therefore, it is obvious that it can not be used to predict RNA secondary structure since it requires inputting the RNA secondary structure for the input sequence (for test).
> > > >
> > > >
> > > > ***we spent efforts on another more reasonable homology model***
> > > >
> > > > We think it possible to use another homology model, e.g., BLAST [2]. We also incorporated the sequence alignment method in [3] to predict the structure. However, we found that for many sequences in the test set, BLAST fails to return a homolog from the training set. Even when we set the E value of the BLAST program as 10, which is expected to return some hits with low qualities, we did not receive any homolog hits for those sequences. That means the homology-based method can not resolve the RNA secondary structure prediction problem alone, even for those known RNA families, whose structures have presented in the database.
> > > >
> > > > However, we are still interested in the overall performance of the alignment-based method on our test set since its performance is usually *missed* in the previous publication, even in those papers about homology-based methods [4]. By combining BLAST, Clustal, and the contact map formulation from E2Efold, we managed to obtain the performance of the alignment-based methods, whose F1 score is 0.79. Indeed, this performance is good, but it’s still worse than E2Efold. Notice that although we transferred some ideas from E2Efold to the alignment-based method to make it work, E2Efold’s performance is still better than the mixed method, which suggests the effectiveness of our framework.
> > > > If you are interested in running this alignment-based method proposed by us, the code is here:
> > > > https://drive.google.com/open?id=19d8nBYQx2qHEtEq-cMsmUkQ8bS9F-UEZ
> > > >
> > > >
> > > > [1] Fallmann, Joerg, et al. "Recent advances in RNA folding." Journal of biotechnology (2017)
> > > > [2] Altschul, Stephen F., et al. "Gapped BLAST and PSI-BLAST: a new generation of protein database search programs." Nucleic acids research (1997)
> > > > [3] Sievers, Fabian, and Desmond G. Higgins. "Clustal Omega, accurate alignment of very large numbers of sequences." Multiple sequence alignment methods. 2014
> > > > [4] Zhen Tan, Yinghan Fu, Gaurav Sharma, David H. Mathews. "TurboFold II: RNA structural alignment and secondary structure prediction informed by multiple homologs." Nucleic Acids Research (2017)

---

> > > > > ### Author Response · Authors · 2019-11-11
> > > > > **Setting 1 is a reasonable, read-world setting and the solution is not trivial**
> > > > >
> > > > > We are happy that Thomas found our method novel, but we can not agree with some of his statements.
> > > > >
> > > > > ***setting 1 is a reasonable and real-world setting***
> > > > >
> > > > > (1) We can not agree with the conclusion that: the comparison is unfair since E2Efold can incorporate information from seen sequences and others can not. The setting is fair as long as in practice we have access to data like that. Especially, the dataset of RNA structures is continually growing.
> > > > >
> > > > > (2) As we mentioned earlier, some methods we compared with use a fitted energy function. We note that these energy functions can be fitted from a much larger set of available datasets that are not used in E2Efold. In some sense, these methods may actually have an advantage over E2Efold in terms of datasets used for training.
> > > > >
> > > > > (3) We do not think traditional models are *designed for* setting 2 as Thomas stated. Instead, they are designed in that way and are commonly adopted because it is more interpretable and biologists care about interpretability. In contrast, deep learning models have better abilities to learn the common pattern from data and make more accurate predictions, while it is less interpretable. (However, DL is a hot topic and now there are many methods to explain the prediction of DL models.)
> > > > >
> > > > > In conclusion, in terms of prediction accuracy, our comparison is fair and predicting structures for unseen sequences is the real-world setting.
> > > > >
> > > > >
> > > > > ***predicting structures for unseen sequences in setting 1 is not a trivial task***
> > > > >
> > > > > Please refer to the results of *CDPfold* in Table3 and the result of *$U_{\theta}$+PP* in Table 6. Both of them are deep learning methods. If it is a trivial task and neural networks only need to memorize and overfit the training data, there won't be such a performance gap between those DL models and E2Efold. They should also work extremely well as long as the neural network is not so small. Especially, *$U_{\theta}$+PP* has exactly the same architecture as E2Efold, except that the gradient is not pushed through the post-process during the training phase.
> > > > >
> > > > > Therefore, designing a suitable deep learning model to better learn the pattern of the structures is not easy. We should not take its performance for granted.

---

### Author Response · Authors · 2019-11-12
**The choice of the number of unrolling iterations T**


We explained the choice of the number of unrolling iterations T in this post, since all reviewers asked questions about this constant.

The performance that we reported in the paper is when T=20. The overall ideas are:
- to ensure that the constraints are mostly satisfied, T can not be too small.
- a very large T will make the neural network very deep and the training process more expensive and unstable.

What we actually did is:
- We first pre-train the score network $U_{\theta}$ without considering the post process network.
- Then we fix this trained $U_{\theta}$ and try different T without further training  (Table 1).
- After the best T is selected, the score network and the post-processing network are trained end-2-end.

Table 1: Constraints-check and validation F1 scores
           | $\max_{A}c(A)$ | $\text{mean}_{A}c(A)$ | validation F1
T = 5   |	      160	     |       28.53      |    0.772
T = 10 |         33          |        7.78       |     0.791
T = 20 | 	       4	     |       0.65        |     0.806
T = 30 |         1	     |       0.03        |     0.809
T = 50 | 	       1	     |       0.03        |     0.808

In Table 1,
- $c(A):=\text{sum}(\text{relu}(A\mathbf{1} - \mathbf{1}))$ is measuring whether the constraints are satisfied. The smaller the better.
- validation F1 scores for T=20-50 are similar. This motivates us to choose T=20 for more efficient training.


In addition, we’ve also tried to train the model for T=50 end-to-end.
Table 2: T=20 vs T=50
           | time per gradient step | best training loss | best validation loss | validation F1
T = 20 | 	         0.64s	               |	       0.63	          |	       0.64                   |    0.88
T = 50 | 	         0.98s	               |	       0.65            |           0.66	                 |    0.86

In Table 2:
- The training time is evaluated on the Titan Xp card with batch size = 8. (When the batch size is 16, it will be out of memory for the case T=50.)
- The final performances are indeed similar. It is not that sensitive to the choice of T.
- The performance for T=50 is a bit worse, which might be caused by the depth of the network, making the optimization harder.

---

### Author Response · Authors · 2019-11-12
**Summary of major revisions**

We would like to thank all the reviewers for their careful reading, detailed comments, and overall positive assessment. We have responded to every raised question and concern, and incorporated your constructive suggestions into the revised version. The major revisions are summarized:


***More Details***

We’ve improved the clarity of the paper by adding
- More related works in Section 2 and Appendix A.
- Appendix B: Derivation Of The Proximal Gradient Step
- Appendix C: Implementation And Training Details
We will also release our code for others to reproduce all the experimental results.


***More Experimental Results***

We’ve included additional results of experiments suggested by both the reviewers and the public comments.
- Appendix D.2. Two-sample Hypothesis Testing. (For better understanding data distributions in RNAStralign and ArchiveII)
- Appendix D.3. Performance On Long Sequences: Weighted F1 Score
- Appendix D.4. ArchiveII Results After Domain Sequences Are Removed
- Appendix D.5. Per-family Performances


***Flow of the paper***

As suggested, we add a new section “RNA Secondary Structure Prediction Problem” which follows the Related Work Section to formally state the problem and concrete constraints earlier than the previous version.


Thank all the reviewers for the constructive suggestions on paper refinement!

---

### Decision · Program_Chairs · 2019-12-19

**Decision:**

Accept (Talk)

**Comment:**

This paper proposes a RNA structure prediction algorithm based on an unrolled inference algorithm. The proposed approach overcomes limitations of previous methods, such as dynamic programming (which does not work for molecular configurations that do not factorize), or energy-based models (which require a minimization step, e.g. by using MCMC to traverse the energy landscape and find minima).

Reviewers agreed that the method presented here is novel on this application domain, has excellent empirical evaluation setup with strong numerical results, and has the potential to be of interest to the wider deep learning community. The AC shares these views and recommends an enthusiastic acceptance.